

# Multi-year black carbon observations and modeling close to the largest gas flaring and wildfire regions (Western Siberian Arctic)

**Olga B. Popovicheva[1], Marina A. Chichaeva[2], Nikolaos Evangeliou[3,\*], Sabine Eckhardt[3], Evangelia Diapouli[4], and Nikolay S. Kasimov[2]**

[1]SINP, Lomonosov Moscow State University, 119991 Moscow, Russia

[2] Faculty of Geography, Lomonosov Moscow State University, 119991 Moscow, Russia

[3]NILU, Department for Atmospheric & Climate Research (ATMOS), 2007 Kjeller, Norway

[4]ERL, Institute of Nuclear and Radiological Science & Technology, Energy & Safety, NCSR Demokritos, 15341 Attiki, Athens, Greece

\* Corresponding author: N. Evangeliou (Nikolaos.Evangeliou@nilu.no)



## Abstract

The influence of aerosols on the Arctic system remains associated with significant uncertainties, particularly concerning black carbon (BC). The polar aerosol station "Island Bely", located on Bely Island (Kara Sea) in the Western Siberian Arctic, was established to enhance aerosol monitoring in the Arctic. Continuous in-situ measurements from 2019 to 2022 revealed the long-term effects of light-absorbing carbon. During the cold period, the annual average light absorption coefficient was $0.7 \pm 0.7$ Mm$^{-1}$, decreasing by approximately 2-3 times during the warm period. The interannual mean showed a peak in February ($0.9 \pm 0.8$ Mm$^{-1}$), a ten times lower minimum in June, and exhibited high variability in August ($0.7 \pm 2.2$ Mm$^{-1}$). The absorption Ångström exponent indicated presence of mixed and aged BC. An increase of up to 1.5 at shorter wavelengths from April to September suggests contribution from light absorbing brown carbon. The annual mean $eBC$ demonstrated considerable interannual variability, with the lowest in 2020 ($24 \pm 29$ ng m$^{-3}$). Significant difference was observed between Arctic haze and Siberian wildfire periods, with record-high pollution levels in February 2022 ($110 \pm 70$ ng m$^{-3}$) and August 2021 ($83 \pm 249$ ng m$^{-3}$). During the cold season, 92% of surface BC was attributed to anthropogenic sources, mainly from gas flaring. In contrast, during the warm period, Siberian wildfires contributed to BC concentrations by 47%. Notably, unprecedented smoke was transported from Yakutian wildfires at high altitudes in August 2021, marking the most severe fire season in the region over the past four decades.



## 1    Introduction

Multiple socio-economic drivers and feedbacks, including air pollution (Arnold et al., 2016)
influence the natural and human environment of the Arctic. Over the last few decades, the Arctic
warms more than three times faster than the global average (AMAP, 2021). The pronounced rapid
changes affect atmospheric transport and aerosol relative source contributions (Heslin-Rees et al.,
2020). Drier conditions and warmer temperatures are the main cause of enhanced fire activity. Boreal
forest fires become more frequent and severe (Rogers et al., 2020), especially in Central Siberia, and
Northern America (Kasischke and Turetsky, 2006; Kharuk and Ponomarev, 2017; Veraverbeke et al.,
2017). Widespread smoke plumes, particularly in Siberia, lead to substantial deterioration of air
quality increasing fine particulate matter (Silver et al., 2024).
Complicated processes affect Arctic pollution and its climate impacts (Willis et al., 2018), such
as the "Arctic haze" in winter and spring. To understand such phenomena and thus reduce their
impact, there is a clear need for comprehensive studies of the climate-relevant aerosol processes that
occur in the Arctic. A species of major concern is black carbon (BC), a short-lived climate forcer
(Schmale et al., 2021). BC is emitted from the incomplete combustion of fossil fuel and biomass; it
is defined as the portion of carbonaceous aerosols, which absorb strongly in the entire climate relevant
wavelength region of the solar spectrum (i.e. IR-VIS-UV). BC contributes to Arctic warming in
multiple ways (e.g., Lee et al., 2013), including the darkening effect of BC deposited on snow and
ice (Flanner, 2013). AMAP (2015) reports that the Arctic equilibrium temperature response is
(+0.4∘C) due to forcing from atmospheric BC and (+0.22∘C) due to snow BC.
At present, the largest uncertainties when assessing aerosol impact on the climate are attributed
to BC (AMAP, 2021). To follow-up on this, BC measurements are taken at various polar regions in
the European, Siberian, and Canadian Arctic (Stone et al., 2014; Yttri et al., 2014; Popovicheva et al.,
2019a; Winiger et al., 2019; Manousakas et al., 2020; Gilardoni et al., 2023). For instance,
Stathopoulos et al. (2021) reported on the long-term impact of light-absorbing carbon in the high
Arctic by analysing 15 years of data from the Zeppelin station (Svalbard), while Schmale et al. (2022)
studied the status of the Arctic haze peak concentrations at 10 Arctic observatories.
There is a large diversity in magnitude and variability of aerosol optical properties, reflecting
differences in sources throughout the Arctic (Schmeisser et al., 2018). The most complicated issue
for BC measurements is various instrumentations and methods that increase uncertainty (Sharma et
al., 2017; Asmi et al., 2021; Ohata et al., 2021). The optical properties of BC have been previously
evaluated against direct mass measurements techniques (Sharma et al., 2004; Eleftheriadis et al.,
2009; Yttri et al., 2024). The conversion of light attenuation to absorbing carbon mass concentration
is performed by the mass-specific absorption coefficient ($MAC$) (Petzold et al., 2013) that is highly





influenced by the aerosol mixing state and non-BC light-absorbing species such as organic matter
and mineral dust (Zanatta et al., 2018) and varies in time and space depending on sources and
transformations during transport (Bond et al., 2013; Chen et al., 2023). It is crucial to quantify the
contribution of non-BC component and aging in order to determine the actual $MAC$ value
experimentally at each site (Singh et al., 2024).

The absorption Angstrom exponent, $AAE$, defined as the relative fraction of wavelength -

dependence of absorption of BC versus other light absorbing constituents, also differs from site to
site (Schmeisser et al., 2018). A fraction of organic aerosol, named brown carbon (BrC), increases
the aerosol absorbing properties at short UV-VIS wavelengths (Sandradewi et al., 2008; Grange et
al., 2020; Helin et al., 2021) and dominates the absorption during wildfire seasons (Bali et al., 2024).
BrC originates mainly from biomass burning and can impose strong warming effect in the Arctic,
especially in the summertime (Yue et al., 2022).

Despite its remoteness, the Arctic is one of the main receptors of anthropogenic air pollutant

emissions from the Northern Hemisphere (Stohl et al., 2013). BC trends and seasonality at three
Arctic sites, Alert (Canadian Arctic), Barrow (American Arctic), and Zeppelin, Ny-Ålesund
(European Arctic) reveal a negative trend of 40% over 16 years due to the anthropogenic emission
reduction (Sharma et al., 2013). The recent increase in fires and their earlier starts, due to the ongoing
warming, have made wildfires in the Northern Eurasia a significant source of Arctic BC (Evangeliou
et al., 2016). Fossil fuel combustion is the major source of BC in the Arctic troposphere (50–94%)
(55–68% at the surface and 58–69% in the snow) and biomass burning (BB) dominates at certain
altitudes (600–800 hPa) between April to September (Qi and Wang, 2019).

Northern Eurasia, particularly Siberia, in a key source region of pollution in the Arctic. Source

quantification (Zhu et al., 2020) shows that surface Arctic BC originates mainly from anthropogenic
emissions in Russia (56%). The reason for this is that the largest oil and gas producing facilities of
Western Siberia are located along the main pathway of air masses that enter the Arctic and thus have
a disproportionally large contribution to the Arctic lower troposphere (Stohl, 2006; Stohl et al., 2013).
Eleftheriadis et al. (2009) and Tunved et al. (2013) identified these regions as a key source for the
highest measured BC in the European Arctic. The impact of long-range transport from these regions
has been previously reported in Ice Cape Baranova station (Manousakas et al., 2020) and Tiksi
(Northeastern Siberia) (Winiger et al., 2017; Popovicheva et al., 2019a). Airborne observations over
the coast of the Arctic seas have identified the long-term transport of the industrial pollution (Zenkova
et al., 2022). Furthermore, efforts have sought to develop BC emission inventories for the Siberian
Arctic, based on activity data from local information, improved gas flaring emissions, and satellite
data (Huang et al., 2015; Böttcher et al., 2021; Kostrykin et al., 2021; Vinogradova and Ivanova,



2023). To better quantify the source contribution to the Arctic environment, targeted aerosol measurements close to the flaring facilities are needed. The present operating Eurasian Arctic stations are all too far away to allow assessing how air masses are affected by different source categories (Stohl et al., 2013). Though, episodic observations of BC at the proximity of the flaring regions have provided a better constraint (Popovicheva et al., 2017b).

Another major source of the Arctic BC is wildfires in the Siberian and Far Eastern regions, which have grown in recent summers (Bondur et al., 2020). Airborne observations of BC in Siberia have confirmed impact forest fires (Paris et al., 2009). Eastern Siberia (Yakutia) has been prone to large wildfires due to a combination of hot summers (> 40ºC) and low humidity (Tomshin and Solovyev, 2022). For instance, wildfires in summer 2019 in Eastern Siberia occurred along the trans-Arctic transport pathway resulting in enhanced aerosol load observed in Western Canada (Johnson et al., 2021). BB emissions occurring at midlatitudes reached the European Arctic in 2020 influencing aerosol composition (Gramlich et al., 2024).

Despite the necessity for detailed observations in the Northwestern Siberia, a dense observational network is still absent. Towards this, the polar aerosol station on the Bely Island (Kara Sea, Western Siberia) started to operate in August 2019 (Popovicheva et al., 2022, 2023). The significance of high-quality measurements at the "Island Bely" Station (hereafter "IBS") is documented, as the station is located along the main pathway of large-scale emission plumes from industrial regions and Siberian wildfires entering the Arctic. Further investigation performed at IBS in August 2021 showed impact from a long-range transport event with unprecedented high concentrations of carbonaceous aerosol (Schneider et al., 2024).

In this paper, we show improved light absorption long-term measurements and BC seasonal and inter-annual variability in the Western Siberian Arctic from three and a half years (2019-2022) of observations at IBS. BC was calculated in two ways: as equivalent BC ($eBC$) by an aethalometer and as elemental carbon (EC) by thermal-optical analysis. We further evaluate the seasonal changes in the observed absorption coefficients. Seasonal difference in intensive optical properties is shown by the wavelength-dependent absorption Ångström exponent ($AAE$), which acts as indication of the BrC impact. Estimated site-specific absorption coefficient ($SAC$) considered the specific seasonal effects of mixing and aging of aerosols at IBS. We further assess the inter-annual variability of origin, transport and main BC sources using modelling tools coupled with the most recent anthropogenic and biomass burning emission datasets.



## 2   Methods

### 2.1   Polar aerosol "Island Bely" station (IBS), location and meteorology

The aerosol "Island Bely" station (IBS) of Moscow State University (73020'7.57"N, 70040'49.05"E) is shown in **Figure *1*a** together with other Polar Arctic observatories. Western Siberia is the world's largest gas flaring region with a leading oil and gas production industry (**Figure *1*b**). It is also an area under intensive exposure by Siberian wildfires (Tomshin and Solovyev, 2022; Voronova et al., 2022).

The climate at IBS is characterized by a large annual variability determined by alternating periods of the polar night and midnight sun. Basic meteorological parameters, such as temperature, wind speed and direction were obtained every 3 hours from a meteorological station located 500 m away from the IBS. The cycles of temperature, precipitation, snow coverage, and wind speed and relative humidity are shown in **Figure S *1***. Annual temperature varied from -39°C to 23°C (mean: -6±12°C) (**Table S *1***). For further analysis, we have split the annual cycle in two periods, November-April ("cold period") and May - October ("warm period"). High relative humidity of 87±8% was typical for the study period, with less than 80% observed in winter 2020. Precipitation was maximum in summer (22 mm) with constant snow coverage from October to May. Wind was relatively stable, with a mean speed of 6±3 m s-1, which increased in winter up to 17 m s-1 (**Figure S *1***).

Wind pattern for the cold period in **Figure *1*d** show a prevailing wind direction from south, southwest, and southeast. Winds were predominantly continental, rarely occurring from the ocean; significant emission sources from the continent were downwind. In the warm period, the wind pattern was more spatially homogeneous with northeastern direction. Period from June and September was characterized by a frequent occurrence of oceanic air masses and constant wind speeds.

The aerosol pavilion takes place approximately half a km to the southeast of the meteorological station. An aerosol sampling system composed from three total suspended particle (TSP) inlets has been installed approximately 1.5 m above the roof and 4 m above the ground. They are equipped with an electric heating wire to prevent rimming and ice blocking of the system. One inlet was used for the real-time light-absorption measurements with air flow 5 L min-1. Two other inlets provided the aerosol sampling by low-volume samplers (Derenda, Germany) operating at 2.3 m3 h-1 flow (0°C, 1013.25 hPa).

### 2.2   Aerosol optical and chemical characterization

An Aethalometer model AE33 (Magee Scientific, Aerosol d.o.o.) was used to measure the light attenuation caused by particles deposited on two filter spots at different flow rates (Drinovec et al., 2015) and at seven wavelengths from ultraviolet (370 nm) to infrared (950 nm). The "dual spot"





technique is applied for real-time loading effect compensation. Optical absorption of aerosols on the
filter is influenced by scattering of light within the filter; the enhancement of optical absorption is
described by the factor C that depends on the filter material. The producer recommends an
enhancement factor of 1.57 for TFE-coated glass fiber filter. The light-absorbing content of
carbonaceous aerosol is reported as equivalent black carbon concentration ($eBC_{AET}$) for the given
wavelength $\lambda$, which is determined for each time interval from the change in the light absorption
using the mass absorption coefficient ($MAC$). The aerosol optical absorption coefficient is therefore:
$$b_{abs}(\lambda) = eBC_{AET}(\lambda) \times MAC(\lambda) \quad (1)$$

where $eBC_{AET}$ at 880 nm is determined using the $MAC$ of 7.7 m2 g-1. The aerosol optical absorption
coefficient for different wavelengths is determined with their $MAC$ values that are equal to 11.58, and
13.14 m2 g-1 at 590, and 520 nm, respectively.
To represent the spectral dependence of the light absorption, the absorption Ångström exponent
($AAE$) was derived by using a fitted power law relationship:
$$b_{abs}(\lambda) = b_{abs}(\lambda_o) \times \left(\frac{\lambda}{\lambda_0}\right)^{-AAE} \quad (2)$$

where $b_{abs}(\lambda_o)$ is the absorption coefficient at the reference wavelength $\lambda_o$, $AAE$ is a measure of
strength of the spectral variation of aerosol light absorption.
BC absorbs strongly in the NIR-VIS with only moderate increment towards the shorter
wavelengths. Light absorbing organic components (BrC) absorb light at shorter wavelengths more
effectively than at 880 nm, which is observed as an increased $AAE$ (Sandradewi et al., 2008; Grange
et al., 2020; Helin et al., 2021). The total light absorption is assumed to include the contribution of
both BC and BrC (Ivančič et al., 2022):
$$b_{abs}(\lambda) = b_{abs/BC}(\lambda) + b_{abs/BrC}(\lambda) \quad (3)$$

Using Eq. 1, the BrC absorption becomes: $b_{abs/BrC}(\lambda) = b_{abs}(\lambda) - b_{abs}(\lambda_0) \times \left(\frac{\lambda}{\lambda_0}\right)^{-AAE}$ (4)
Light-absorption measurements were performed for three and a half years, from 10 August
2019 to 31 December 2022, with a time resolution of 1 min. Data were cleaned based on analysis of
meteorological parameters by examining whether the wind originated from the direction of the
meteorological station where diesel generators operated. In such cases, strong peaks of BC were
removed from further analysis. Around 6.4 % of the hourly-average data were cleaned from the
dataset due to local pollution impact. To avoid the instrumental noise when calculating the $AAE$, the
z-score was used that calculates the ratio of difference between a single raw data value and the data



mean to the data standard deviation. Outliers (< -3 and > 3 of observation's z-score) were removed
from the dataset.

A thermal EC analysis was conducted for the samples in parallel to AE33. Sampling was

performed on 47 mm quartz fiber (Pallflex) filters preheated at 600ºC for 5 h. The low concentrations
of ambient aerosols necessitate that the sampling times reach up to a week, in order to allow the filter
loading to exceed the detection limit for relevant aerosol chemistry analyses. The total number of
samples limited by the low detection limit of the thermal-optical instrument were 180.

Organic (OC) and elemental carbon (EC) were measured by thermo-optical transmittance

(TOT) analysis (Lab OC-EC Aerosol Analyzer, Sunset Laboratory, Inc.) using the methodology
reported in Popovicheva et al. (2019) and Manousakas et al. (2020). Quartz filter samples were heated
first up to 650 °C in He atmosphere and then up to 850 °C in a mixture of 2% O2 in He, using the
controlled heating ramps of the EUSAAR_2 thermal protocol. OC evolves in inert atmosphere, while
the thermal refractory fraction EC is oxidized in the He-O2 atmosphere. Charring correction due to
pyrolytic carbon (PC) was applied by monitoring the sample transmittance throughout the heating
process. The limit of detection (LOD) for the EC analysis was 0.05 µg C cm−2. QA/QC procedures
of EN 16909:2017 were also applied during TOT analysis. Laboratory and field blanks were prepared
and ran following the same analytical procedures as for the samples.

Both methods have important uncertainties (Sharma et al., 2017; Ohata et al., 2021). The

determination of EC by thermo-optical analysis may be impacted by the presence of carbonate carbon
(CC), which is quantified during analysis as OC and/or EC. The contribution of CC in fine aerosol is
generally considered negligible but its interference may be significant for coarse aerosol and samples
heavily impacted by resuspended soil. The split between EC and OC may be also affected by the
presence of light-absorbing species others than EC, such as light absorbing organic carbon. In
addition, the presence of mineral oxides, such as iron oxide, might provide oxygen during analysis
and lead to pre-oxidation of EC in inert atmosphere. $eBC$ might overestimate BC if there are
coexisting components such as BrC (Chakrabarty et al., 2010) and dust (Petzold et al., 2009). In
addition, the aethalometer response depends on filter loading and multiple scattering by the filter
medium and sampled aerosol particles (Backman et al., 2017).

Validations of $eBC$ retrievals were performed against results from thermal-optical analysis of

EC according to an approach that has been used previously in Sharma et al. (2004), Eleftheriadis et
al. (2009) and Yttri et al. (2014). To convert optical absorption at 880 nm to BC mass, the site-specific
mass absorption coefficient ($SAC$) was estimated as:

$$SAC = \frac{b_{abs/BC}}{EC} \qquad (5)$$



Data processing was performed using Deming's total least-squares regression to compare
measurements from different methods and modelling, estimate the mass absorption cross-sections
($MAC$), and evaluate correlations among variables (R package "Deming"; (Therneau, 2024)). Deming
regression fits a couple of variables considering the independent errors of both. The errors are
assumed to be normally distributed; the error ratio is 1, and the regression results are equivalent to
the orthogonal regression with the intercept forced through zero.
**2.3    Atmospheric dispersion modelling**
To investigate the air mass transport and possible origin of BC, the Lagrangian particle
dispersion model FLEXPART version 10.4 was used (Pisso et al., 2019) driven by hourly reanalysis
meteorological fields (ERA5) from the European Centre for Medium-Range Weather Forecasts
(ECMWF) with 137 vertical levels and a horizontal resolution of 0.5°×0.5° (Hersbach et al., 2020).
In FLEXPART, computational particles were released at heights 0 - 100 m from the receptor (IBS)
and tracked backward in time in FLEXPART's "retroplume" mode. Simulations extended over 30
days backward in time, sufficient to include most BC emissions arriving at the station, given a typical
BC lifetime of 1 week (Bond et al., 2013). The tracking includes gravitational settling for spherical
particles, dry and wet deposition of aerosols (Grythe et al., 2017), turbulence (Cassiani et al., 2015),
unresolved mesoscale motions (Stohl et al., 2005), and deep convection (Forster et al., 2007). The
FLEXPART output consists of a footprint emission sensitivity that expresses the probability of any
emission occurring in each grid-cell to reach the receptor. The footprint can be converted to modelled
concentration at the receptor, when coupled with gridded emissions from an emission inventory.
Modelled concentrations can be calculated as a function of the time elapsed since the emission has
occurred (i.e., "age"), which can be shown as "age spectrum", while masks of specific
regions/continents can give the continental contribution to the simulated concentration (i.e.,
"continent spectrum").
The source contribution to receptor BC is calculated by combining each gridded emission sector
(e.g. gas flaring, transportation, waste management etc…) from an emission inventory with the
footprint emission sensitivity (as described in the previous paragraph). Calculations for anthropogenic
sources (emission sectors are described below) and open biomass burning were performed separately.
This enabled identification of the exact origin of BC and allowed for quantification of its source
contribution. Anthropogenic emissions were adopted from the latest version (v6b) of the ECLIPSE
(Evaluating the CLimate and Air Quality ImPacts of ShortlivEd Pollutants) dataset, an upgraded
version of the previous version (Klimont et al., 2017). The inventory includes emissions from
industrial combustion (IND), from the energy production sector (ENE), residential and commercial
emissions (DOM), emissions from waste treatment and disposal sector (WST), transportation (TRA),



shipping activities (SHP) and gas flaring emissions (FLR). The methodology for obtaining emissions
from FLR specifically over the Russian territories has been improved in ECLIPSEv6 (Böttcher et al.,
2021). Annual total and monthly anthropogenic emissions are shown in **Figure S 2**. Biomass burning
was adopted from the Copernicus Global Fire Assimilated System (CAMS GFAS) (Kaiser et al.,
2012) because this product provides an estimation of the injection altitude of the fire emissions that
is crucial for accurate simulation of the BB dispersion. Annual total and daily fire emissions from
CAMS GFAS are shown in **Figure S 3**. A satellite image of smoke plume for 5th August 2021 was
obtained from https://worldview.earthdata.nasa.gov. Fires are shown from the Fire Information for
Resource Management System (FIRMS) (https://firms.modaps.eosdis.nasa.gov/map) ten days back
in time.

## 3    Results

### 3.1    Aerosol light-absorption

Light-absorption coefficients at 880 nm, $b_{abs}(800)$ were used to infer $eBC$ mass
concentrations. $b_{abs}(800)$ were plotted as hourly and monthly means during the entire study period
(2019-2022) (**Figure 2**). **Table 1** presents the data statistical summary. The mean $\pm 1\sigma$ (median) value
of $b_{abs}(880)$ was $0.5\pm0.9$ (0.27) Mm−1 for the entire study period. In the cold period the annual
average mean (median) of $b_{abs}(800)$ was $0.7\pm0.7$ Mm-1 (0.5), during the warm period it was 1.9
(2.5) times less. There is a clear seasonality consistent with the Arctic aerosol light absorption from
other studies (Stathopoulos et al., 2021; Schmale et al., 2022; Pulimeno et al., 2024). 15 years (2001-
2015) record at Zeppelin demonstrated that the long-term seasonality of light absorbing carbon
(Stathopoulos et al., 2021) $b_{abs}(800)$ was 0.112 Mm-1 (median) in the cold period and 0.035 Mm-1
in the warm period; both values approximately 5 times less than those observed at IBS.
Monthly means of $b_{abs}(800)$ for each year together with intra-annual means for IBS are shown
in **Figure 2**. Specifically, annual average $b_{abs}(800)$ exhibits a significant peak during winter and
summer for any year. The examination of the overall changes by the inter-annual mean reveals a
gradual increase from November ($0.4\pm0.5$ Mm-1) to February ($0.9\pm0.8$ Mm-1); the latter represents
the maximum light absorption observed at IBS. In February, the monthly mean of $b_{abs}(800)$ ranged
from 0.4 to 1.7 Mm−1 reaching the maximum (1.7 Mm−1) in 2022. Thus, Arctic haze is present at
IBS in winter months, from December to February. Starting from March ($0.6\pm0.5$ Mm-1), the inter-
annual mean decreased down to a minimum in June ($0.1\pm0.2$ Mm-1) that was 9 times less than that
of February. August had the highest light-absorption (mean: $0.7 \pm 2.2$ Mm-1) within the summer
months, ranging from 0.2 to 1.5 Mm−1 and showing a maximum of 1.5 in 2021. September and



October demonstrated a similar level of variability with June. The annual monthly mean
concentrations for all study years are within ±1σ of inter-annual mean concentrations (**Table *1***) and
include 68% of the observed data. At Zeppelin, the maximum of the intra-annual (2001-2015) mean
was seen in March - April (0.3 Mm-1) (Stathopoulos et al., 2021), coinciding with the Arctic haze
phenomenon in late winter-spring that has been widely observed in the European and Canadian Arctic
(Sharma et al., 2004; Schmale et al., 2022).
In order to relate the light absorption in visible spectrum to the variability on other locations
(Schmeisser et al., 2018; Pulimeno et al., 2024), we calculate $b_{abs}$ at 520 and 590 mn. The mean
(median) value of $b_{abs}(520)$ was 0.9±1.6 (0.5) Mm−1 for the entire study period (**Table *1***). At Ny-
Ålesund (Svalbard), the annual mean (median) $b_{abs}(530)$ averaged for 2018 to 2022 was 0.22 (0.13)
Mm−1 (Pulimeno et al., 2024), approximately 4 times less. Moreover, the absorption coefficient
$b_{abs}(550)$ of 0.18 (0.09) Mm−1 recorded for 2012-2014 again in Svalbard (Schmeisser et al., 2018)
was 4 times less compared to annual average light absorption at IBS.
We present multi-annual box-and-whisker plots of $b_{abs}$ at 590 nm in **Figure *3***. The wavelength
of 590 nm was chosen as the closest to 550 nm reported for the polar station Tiksi (Schmeisser et al.,
2018; Schmale et al., 2022). The monthly medians of $b_{abs}(590)$ in February ranged from 0.3 to 2.3
Mm−1, representing the highest values observed in 2022. The highest extended interquartile range
(up to 1 Mm−1) was observed in the cold period. Conversely, the summer months exhibited a
minimum of approximately 0.1 Mm−1 for $b_{abs}(590)$ with smaller variation of data characterized by
the low interquartile range of 0.4 Mm−1.
The annual cycle of $b_{abs}(590)$ reflects the higher aerosol burden during the haze season and
the low concentrations during summer at Alert, Barrow, Zeppelin, Gruvebadet, and Tiksi (Schmale
et al., 2022). Seasonality of $b_{abs}$ medians at 550 nm for polar stations (Alert, Barrow, Tiksi, Zeppelin)
from (Schmeisser et al., 2018) and $b_{abs}(590)$ for IBS are presented in **Figure *3***. All sites demonstrate
similar seasonal variations, albeit a different magnitude of light absorption. In February, the
maximum $b_{abs}(590)$ (1.1 Mm−1) was observed at IBS; a higher value has been only observed at
Tiksi which is explained by the influence from local sources (Popovicheva et al., 2019a). Other
stations show the Arctic haze maximum later (in March or April); a sharp decline of $b_{abs}(590)$ was
observed at those months at IBS. Values similar to other Arctic stations were recorded at IBS in June,
with an annual minimum of around 0.1 Mm−1. Since July, $b_{abs}(590)$ at IBS was higher than at other
stations except Tiksi and peaked at 0.8 Mm−1 in December. The polar station Pallas exhibits the
opposite behaviour peaking in spring and summer (Schmeisser et al., 2018). Pallas is located
relatively south as compared to the rest of the polar stations and, hence, it is influenced by
anthropogenic and biogenic emissions from surrounding boreal forests (Asmi et al., 2011). Finally,



we conclude that at IBS the aerosol optical properties in the IR and visible solar spectrum are different
from European, Canadian and Western high-latitude polar locations: light absorption coefficients are
higher during the annual cycle as well as Arctic haze is the most prominent in December-February.
**3.2     Black carbon and site-specific mass absorption cross-section**
Elemental carbon (EC) collocated with light absorption observations is widely used to infer BC
(Grange et al., 2020). **Figure 4**a shows concentrations of EC determined for samples collected in
parallel with the aethalometer measurements from 10 August 2019 to 31 December 2022, with
$eBC_{AET}$ concentrations averaged over the sampling period. Both weekly EC and $eBC_{AET}$
concentrations show the same seasonal variations with a maximum in winter and minimum in
summer. EC concentrations are generally smaller than $eBC_{AET}$. The annual EC mean concentrations
ranged from 6.5 to 16.3 ng C m-3. The highest EC (0.2 µg C m-3) was recorded in December 2019
and the highest $eBC_{AET}$ (0.4 µg m-3) in December 2019 and January 2022. EC was higher (0.05±0.03
µg C m-3) in the cold period and decreased (0.02±0.03 µg C m-3) in the warm period (**Table 1**).
Annual average mean EC during the entire study period was 0.03±0.03 µg C m-3. For comparison,
at Zeppelin and Villum the annual mean EC concentrations were 0.012 ±0.04 µg C m-3 (2017-2020)
(Yttri et al., 2024) and 0.029 ±0.03 µg C m-3 (2011-2013) (Massling et al., 2015), respectively.
Annual mean OC concentrations during the entire study period were estimated as 0.45±0.3 µg
C m-3. At Zeppelin, annual OC (2017-2020) was 3.5 smaller (0.13±0.1 µg C m-3) (Yttri et al., 2024).
Notably, the multi-year average EC and OC levels at IBS are approximately 3 times higher than at
Zeppelin, that correlates well with increased light absorption, as described previously. At IBS, OC
was 0.4±0.2 µg C m-3 in the cold period and increased to 0.5±0.4 µg C m-3 in warm period, opposite
to EC (**Table 1**). The ratio OC/EC shows increased OC and decreased EC in the warm period and an
opposite tend in the cold (**Figure 4**b). **Figure 4**c depicts the relationship between $eBC_{AET}$ and EC in
cold and warm periods. We note the high R2 values for the cold period (0.88) and slightly lower ones
for the warm one (0.78). During the warm period, seasonal mean values reveal an overestimation of
$eBC_{AET}$ that is more pronounced during the warm period, with a slope equal to 2.3. R2 values were
lower because many EC values were close to the LOD. Seasonal differences are attributed to pollutant
sources altering the chemical composition of aerosol at IBS. A positive correlation was observed
between $eBC_{AET}$/EC and OC/EC indicating that BC at IBS is coated with OC leading to the lens effect
(Kanaya et al., 2008) and overestimating $eBC$.
Similar seasonal variation for $eBC$ and EC with highest winter and lower summer
concentrations has been observed previously at Villum, with a regression slope of 2 and a R2 of 0.64
(Massling et al., 2015). At Alert, the median $SAC$ during the Arctic haze season (November to April)
was 19.8 m2 g-1 (Sharma et al., 2004). However, during the non-Arctic haze period from May to



October it was significantly higher 28.8 m2 g-1 and much more variable. This is explained by aged,
internally mixed, and of anthropogenic origin of winter and spring arctic aerosols while summer
aerosols were affected by local sources.
Following the definition in Eq.5, we calculate the $SAC$ from the slope of BC light absorption at
880 nm, $b_{abs/BC}(880)$, and EC concentrations. $SAC_{BC,cold}$ (for the cold period) was estimated to be
15.9 m2 g-1 while $SAC_{BC,warm}$ was higher (18.1 m2 g-1) (**Figure 5**). $SAC$ values at Alert have been
reported to be even higher (Sharma et al., 2004), showing that Western Arctic aerosols differ by
composition and aging. Recalculations of BC mass with $SAC$ values for cold and warm periods ($eBC$),
separately, were performed according to Eq.1.
Timeseries of daily and monthly mean $eBC$ concentrations from August 2019 to 31 December
2022 are shown in **Figure 2**. Annual mean and median $eBC$ for the entire period were 28.7 ± 54.1
ng/m³ and 12.5 ng m⁻³, respectively (**Table 1**); they exhibit a strong year-by-year variability. $eBC$
climatology and the statistics for each month and year of study are presented in **Figure 2** and **Table**
**S 2**. The annual mean $eBC$ in 2019, 2021 and 2022 was 33±44, 33±85, and 32±48 ng m⁻³, respectively.
Statistically significant difference at the 95% confidence level with a p-value <0.05 (t-test) was
observed for the cold and warm periods with means 44±47 and 19±57 ng m⁻³, respectively. The
smallest mean $eBC$ of 24±29 ng m⁻³ occurred in 2020. The latter is likely attributed to the impact of
COVID-19 restriction measures to the emissions of BC (Evangeliou et al., 2020).
The general trend of the maximum in winter and minimum in summer well reproduces the
typical $eBC$ seasonality reported in polar observatories (Stone et al., 2014; Schmale et al., 2022).
**Figure 2** shows monthly mean $eBC$ concentrations for half of year 2019 and whole - year periods of
2020, 2021, and 2022 as well as annual averaged monthly mean $eBC$ climatology for the entire study
period. The highest concentration in the cold period was observed in December 2019 (82±67 ng m⁻³),
January 2022 (63 ±51 ng m⁻³), February 2022 (106±67 ng m⁻³), and March 2021 (32±24 ng m⁻³)
(**Table S 2**). In warm periods we recorded the highest concentrations in September 2020 (31± 48 ng
m⁻³), August 2021 (83±249 ng m⁻³), April 2021 (35± 26 ng m⁻³), and August 2022 (28±54 ng m⁻³).
**3.3    Multi-wavelength absorption Angstrom exponent**
As shown by Virkkula (2021), pure BC particles surrounded by non-absorbing coatings can
have absorption Angstrom exponent ($AAE$) in the range from <1 to 1.7. Compendium of values from
different emissions show $AAE$ variation from 0.2 to 3.0 for transport, power plants, and domestic
wood burning (Helin et al., 2021). Primary emissions from residential heating (Cuesta-Mosquera et
al., 2024) and biomass burning (Popovicheva et al., 2017a, 2019b) have been associated with high
$AAE$ of around 3-4. Due to the mixing with background aerosol, coating and aging processes, a large



change in the light absorption has been reported at receptors of long-range transported pollution
(Cappa et al., 2016). For highly aged aerosols, *AAE* has been found lower than 1.0 due to large and
internally mixed particles (Popovicheva et al., 2022). Spectral absorption was obtained at IBS in the
UV to IR spectral region emphasized by the value of $AAE_{350/950}$ equal to 0.96 for the entire study
period (**Figure 6**a). Power law fittings of spectral dependence for both and cold periods show similar
values, indicating highly mixed and aged BC.

Multiple studies have addressed the sensitivity of the *AAE* to the range of wavelengths

selected for its calculation (Cuesta-Mosquera et al., 2024); the extent of this sensitivity is higher for
aerosols containing a substantial contribution of organic species such as BrC. Events affected by
regional fire emissions were evident by the light absorption coefficient $AAE_{370/520}$ in the short
wavelength range (Ulevicius et al., 2010). In remote Arctic environments, cases with exceeded
$AAE_{467/660}$ have been identified to be influenced by BB (Pulimeno et al., 2024). Impact of intensive
wildfires in North America on aerosol optical properties measured at the European Arctic has been
associated with increased daily $AAE_{467/660}$ of up to 1.4 (Markowicz et al., 2016). Strong UV
absorption has led to increase of up to 1.8, clearly indicating the importance of non-BC light-
absorbing component (Ran et al., 2016).

To apportion the wavelength-dependent light absorption, we used a pair of wavelengths in the

whole wavelength (350 and 950 nm) range, in shorter (370 and 660 nm) and (370 and 520 nm) ranges.
Timeseries of weekly average $AAE_{370/520}$ showed a similar seasonality but wider variation (0.2-3.1)
than (0.5-1.7) for $AAE_{370/950}$ (**Figure 6**b). The mean values increased from 0.97 ± 0.23 for
$AAE_{370/950}$ to 1.17 ± 0.5 for $AAE_{370/520}$ for the entire study period (**Table 1**). Box-whisker plots and
annual averaged means of $AAE_{370/950}$ showed no prominent monthly dependence (**Figure 4**c).
However, increased $AAE_{370/950}$ above 1.1 was observed in summer months for several years, in July
2020, June 2021 and from May to September 2022 (**Table S 2**). The shorter the wavelength pair, the
higher the annual average *AAE* above 1.0. The largest values of monthly mean (median) $AAE_{370/520}$
were found for April to September with a maximum in June. Such considerable deviation during
warm months implies the importance of BrC light-absorbing components within highly mixed Arctic
aerosols at IBS.

Light absorption at 370 nm, $b_{abs}(370)$, was used to estimate the BrC mass concentrations.

The mean (median) value of $b_{abs}(370)$ was 2.4 times higher than $b_{abs}(880)$ for the entire study
period as well as for cold and warm ones (**Table 1**). Monthly means and box-whisker plot of
$b_{abs}(370)$ showed trends similar to $b_{abs}(880)$ (**Figure S 3**). Assuming that the wavelength pair λ
and $λ_0$ in Eq. 3 being 370 and 950 nm, respectively, the absorption coefficient for BrC at 370 nm,




$b_{abs/BrC}(370)$, is determined by subtracting BC absorption from the total absorption at the same
wavelength using the $AAE_{370/950}$ value for entire period (**Table 1**). Monthly $b_{abs/BC}(370)$ and
$b_{abs/BrC}(370)$ as well as the $b_{abs/BrC}(370)$ contribution to total $b_{abs}(370)$ are shown in **Table S 3**
for those years when the contribution of BrC absorption was higher than 1%. We note 13% for August
2021 for the warm period and 5 % for February 2022 and December 2021 for the cold period.

### 3.4 Modelled concentrations of BC

**Figure 7**a shows the monthly mean $eBC$ and surface BC ($BC_{FLEXPART}$) concentrations
simulated with FLEXPART coupled to ECLIPSEv6-GFAS emissions for the entire study period.
FLEXPART model performs well in capturing the seasonality of observed features with both high
and low concentrations. Annual mean modelled BC (88.4 ng m-3) is 37% higher than aethalometer-
measured $eBC_{AET}$ (64.3 ng m-3) and 3 times higher than $eBC$ (29.5 ng m-3). Annual and monthly
means of $eBC_{AET}$ show values closer to $BC_{FLEXPART}$ than $eBC$. This is a reasonable finding because
the global emission datasets could not consider local pollution. Almost all simulated BC
concentrations, except in February 2020 and 2021, were found within the standard deviation range of
measured $eBC_{AET}$. A good correlation between measurements and simulations, with a Pearson
coefficient of 0.72 and 0.7 and a root mean squared error (RMSE) of 15 ng m-3 and 0.14 ng m-3 was
obtained for the cold and warm period, respectively (**Figure 7**b,c).
FLEXPART does not reproduces seasonal variations of BC everywhere over the Arctic. R2
and RMSE varied between 0.53-0.80 and 15.1-56.8 ng m-3, respectively, depending on the location
(Zhu et al., 2020). At Zeppelin, modelled BC (annual mean of 39.1 ng/m3) was reported to be 85%
higher than the measured value (21.1 ng m-3 for annual mean). At Tiksi, modelled BC was
underestimated (74.4 ng m-3 for annual mean) by 40% compared with observations (104.2 ng/m3 for
annual mean) (Zhu et al., 2020). Such good result for IBS is due to its closer location to the biggest
emission sources.
**Figure 8** shows the vertical distribution of simulated BC as a function of time for 2019-2020
years (vertical cross-section). Consistently high vertical BC profiles up to 2 km were observed in the
cold period, except in April 2022. In February 2020, a smoke layer of BC concentrations of up to 100
ng m-3 was prominent at up to 4 km. On the contrary, in the warm period the smoke resides near the
surface, despite a few events of extremely high vertical BC at altitudes up to 8 km and 10 km, which
occurred in July 2020 and August 2021, respectively. Nevertheless, the evidence of atmospheric
transport from high altitudes during summer months is evident by the elevated modelled BC (>100
ng m-3) at high model layers (e.g., July 2019, June-August 2020, June-July 2021 and May-June
2022). In all these periods, surface modelled BC (violet line in **Figure 8**) was under 40 ng m-3
showing that the emission sources are probably far away, and that long-range transport occurred. The





low injection altitude of anthropogenic emissions in winter months cause emitted substances to
remain close to the emission sources. BC climatology at IBS indicates that the long-range transported
anthropogenic emissions in the cold period reside at altitudes up to 2 km and compose a persistent
layer (**Figure 8**). This is further explained by the rapid (about 4 days, or less) low-level transport of
air masses to the Arctic troposphere as described in Stohl (2006). However, this cannot be confirmed
without targeted high altitude observations.

## 4   Discussion


### 4.1   Long-range transport, age and region contributions


Transport mechanisms from the source regions affect the Arctic BC variability and burden

(Chen et al., 2023; Zhou et al., 2012). Transport of aerosols to the Arctic leads to high concentrations
of BC in winter and spring (Arctic haze) and low values in summer (Law and Stohl, 2007) when the
removal processes in the dry and stable Arctic atmosphere are very slow. Synoptic-scale circulation
effects promote the effective transport from lower latitudes, namely diabatic cooling of air masses
moving over snow-covered ground, high continental pressure in winter, and the intrusion of warm air
from lower latitudes (Gilardoni et al., 2023). Seasonal trends of footprint emission sensitivity
demonstrate the transport mechanisms from the source regions to the European Arctic (Platt et al.,
2021). BC at Zeppelin is affected by significantly different source regions during the warm and cold
seasons, while large-scale circulation patterns that affect the pollutant transport from lower latitudes
show the opposite behaviour during these two periods (Stathopoulos et al., 2021).

**Figure 9** shows a 3.5-year climatology of the surface footprint emission sensitivities at IBS.

From December to February, anthropogenic polluted air mass transportation takes place from Eurasia
(territories above 40ºN), as illustrated by the elevated footprints there. The extension of the Arctic
front towards lower latitudes during the cold period facilitates such transport (Stohl, 2006). The
warmer it gets in spring, the narrower the area of emission transport. In the transition from spring to
summer, transport patterns and meteorological conditions change, such as that the advection of the
particulate pollution to the Arctic boundary layer from lower latitudes becomes limited (Bozem et al.,
2019). In JJA (June, July, August) footprint is mostly restricted to coastal regions of Eurasia,
Greenland, and North America and does not extend deeply into the continents. This is a consequence
of the so-called 'polar dome' that prevents warm continental air masses from entering the Arctic
lower troposphere (Stohl, 2006). As a result, anthropogenic pollution becomes less significant, and
natural aerosol sources prevail (Moschos et al., 2022b, a). In autumn (September, October,
November), footprint is similar to the MAM (March, April, May) one completing the annual cycle.





For the entire study period, the monthly mean contribution to surface BC for all years was from air masses with 1-3 (31%) and 3-6 days (22%) aging (**Table S 4**). The highest BC contribution (34%) and (39%) was observed for the shortest age of 1-3 days in DJF (December, January, February) and MAM, respectively (**Figure 8**). In summer, the highest BC contribution (35%) was replaced by a longer age of 6-9 days.

Footprint emission sensitivities of Arctic air masses also constrain the region contributions. The major source regions contributing BC to IBS are the territory of the Russian Federation (including European part of Russia (EURus), Siberia, Far East), Asia, Europe, Northern America, and Ocean. Due to the geographical proximity, EURus/Siberia/Far East contribution (77%) dominated during the entire study period on a basis of the annual average monthly means (**Table S 4**), with a maximum of 83% in SON (**Figure 9**). Its monthly maximum (88%) was recorded in September 2021, and the minimum (60%) in June 2022. Europe was the second region contributor (11%) followed by Asia. The monthly mean contribution of Northern America was up to 12% in JJA, the largest was observed in July 2022 (62%).

### 4.2 Anthropogenic and biomass burning sources

The time series of monthly mean and annual average monthly mean source contributions to surface BC at IBS are shown in **Figure 10**a. Anthropogenic sources (DOM, TRA, IND, FLR, All others) contribute 83% of the total for the entire study period (**Table S 4**). A decrease from winter to July and an increase from August to winter were seen. In the cold period, air masses arrived at IBS through the populated regions of Western Europe, European part of Russia, Siberia, and Asia, crossing the biggest oil and gas extraction regions of Kazakhstan, Volga-Ural, Komi, Nenets, and Western Siberia (**Figure 1**). Because IBS is located north of the largest oil and gas producing regions of Western Siberia, high FLR contribution of 50% and 32% was observed both in the cold and warm period (**Table S 4**). Annual mean contributions to modelled surface BC from FLR, DOM, TRA, and IND sectors dominated in January and December (60%, 22%, 12%, and 9%, respectively). All other sources were around 2% at that time. BB played the biggest role between April (8%) and October (17%), with maximum in August (80%).

**Figure 10**b shows the percentage sectoral contributions on monthly mean BC concentrations for each year. February 2021 and December 2021 were the leaders of FLR impact with 67.2% and 67.4%, respectively. During February 2022 of the record high BC pollution level observed at IBS, air masses arrived at IBS through the Western Europe, European part of Russia, and Siberia, passing through the flaring facilities of Kazakhstan, Volga-Ural, Komi, Nenets, and Western Siberia. They caused of 50%, 26%, 15%, 8%, 0.2%, and 3.3% monthly average contribution to surface BC from FLR, DOM, TRA, IND, BB and All other sources, respectively. Footprint emission sensitivities on



3rd February 2022 at 12:00-15:00 when *eBC* reached 310 ng m-3 (**Figure S 5**) showed air mass
transport to IBS straight through the Western Siberian gas flaring region (**Figure 10**c).

The contribution of FLR dropped significantly from April to a minimum of 18% in June and

rose in September. In the winter months when the overestimation of modelled BC concentrations was
recorded (see section 3.4), the highest FLR impact was seen. DOM showed the biggest contribution
(18%) from November to February, exactly during the heating season. The light absorption of BrC
was significant mostly in wintertime (**Table S 3**). The latter indicates significant impact of biomass
used for domestic heating, in accordance to wood burning contribution of 61% of the total residential
emissions in forest regions (Huang et al., 2015).

According to CAMS GFAS (**Figure S 3**), significant global fire emissions started from June

and lasted until the mid of November in 2020 and 2022; the period of fire emissions was shorter but
more intensive from July until September 2021. At IBS, the annual mean BB contribution approached
48% of the total in the warm season (**Table S 4**). It started increasing from April and approached a
maximum of 80% in August, whereas TRA, DOM, IND, and All other sources were minimum. From
middle June to September, the average monthly BB contribution was larger than all anthropogenic
sources. Notably, from April to September, the high mean BB contribution was related to the excess
of $AAE_{370/520}$ over 1.0 (maximum: 1.7 in July) (**Figure 6**). At that time, the air masses transported
to IBS were aged (> 6 days) dominating the age spectrum (60%) (**Table S 4**).

In 2019, 72.4 thousand km2 were burned in Siberia or 42% of the total burned area that

occurred in Russia (Voronova et al., 2020). A significant relationship between the burned areas and
associated pyrogenic emissions with atmospheric blocking events was reported (Mokhov et al., 2020).
August and September showed 50% and 35%, respectively, monthly mean BB contributions, while
October and November lower, 30% and 20%, respectively (**Figure 10**b).

In spring 2020, BB BC concentrations simulated with WRF-Chem were distributed in areas

between 40ºN and 60ºN in Europe, central Siberia, and East Asia, and indicated intensive seasonal
agriculture fires in Europe and Siberia (Chen et al., 2023). Spring fires contributed about 12% BB
BC to IBS (April and May). The end of June and beginning of July of 2020 was characterized by high
altitude BC (**Figure 8**) indicating high altitude long-range transport. A high BrC content was also
observed in July and September 2020 (**Table S 3**).

In 2021, the monthly mean spring BB contribution approached a maximum of 36% in May.

Yakutia (Eastern Siberia) experienced the worst fire season over the last four decades (Tomshin and
Solovyev, 2022). Around 150,000 occurred, almost twice as much as the previous year (Voronova et
al., 2022). August 2021 received 90% contribution from BB as compared to all the other sources. At
that time unprecedented high smoke levels were recorded over Western Siberia (Schneider et al.,
2024). Satellite image reveals the strong plume from the area of Yakutian wildfires which brought





deep smoke to IBS located around 2000 km far away (**Figure 1**c). The highest *eBC* level of 1800 ng
m-3 on 5th August, exceeded the 75$^{th}$ percentile of the entire period 53 times (**Table 1**)! The measured
concentrations were 180 times higher than the Arctic background (**Figure S 6**). Severe smoke affected
the visibility near IBS (**Figure 1**d). Footprint emission sensitivity on 5th August (from 18:00 to 21:00)
at the time when *eBC* peaked (1540 ng m-3) confirms that air masses originated from Yakutia and
arrived to IBS from the northeast direction (**Figure 10**c). BC for these wildfires was transported at
altitude as high as 10 km (**Figure 8**). Finally, in summer 2022, wildfires took place in Western Siberia
and the European part of the Russian Federation (Popovicheva et al., 2023); BB contributions in June,
July, August 2022 were around 65%, whereas light absorption of BrC was important in May and
August 2022 (**Table S 3**).

## 5    Conclusions

Almost four years (2019-2022) of observations at the aerosol station IBS highlight the light-
absorption characteristics of Western Siberian polar aerosols and its basic cycles, such as seasonality,
annual means, and interannual variability. The annual cycle of multi-wavelength light absorption
demonstrates higher levels during the Arctic haze season and lower ones in summer, similar to other
observatories across the Arctic. The light absorption coefficient revealed a number of unique features:
• Higher magnitude (around 4-5 times) in comparison with multi-year observations at high-latitude
polar stations in European Arctic (annual mean of 0.7±0.7 Mm-1 in the cold season and 2 times
lower in warm).
• Wintertime maximum was observed in February (0.9±0.8 Mm-1) that coincides with the Arctic
haze peak; this is different from the European and Canadian Arctic that is usually observed in
early spring. The interannual minimum was observed in June whereas August was highly
variable with respect to light-absorption due to the Siberian wildfires.
• Multi-annual monthly means for light absorption coefficients in the visible spectrum at IBS were
found higher than at European, Canadian and Western high-latitude polar locations, due to that
IBS is closer to the main Northern Eurasian source regions.
• Wildfire caused increased concentrations, usually in August. Increase of absorption Angstrom
exponent in the UV spectrum between April and September implies coexistence of highly
mixed/aged BC and light-absorbing BrC components, revealing BB aerosols at IBS. Specifically,
monthly BrC contribution to total light absorption was 5 % in February 2022 and 13% August
2021 likely due to wildfire impact.
• BrC light absorption coefficient in the UV spectrum showed similar trends as BC, although it
exceeded BC by 2.4 times during both cold and warm periods. AAE was equal to 0.96, indicating



highly mixed and aged aerosols. AAE in UV spectrum increase up to 1.17±0.5 implies
coexistence of light-absorbing BrC components in biomass burning aerosols, with the biggest
impact between April and September.
• We calculated site-specific mass absorption coefficient ($SAC$) for the first time at IBS by
combining multi-year optical absorption and EC data. Higher $SAC$ of 18.1 m2 g-1 in the warm
period than in the cold one (15.9 m2 g-1) revealed influence from non-BC light-absorbing
species, such as organic matter and mineral dust; $SAC$ values were lower than those observed in
the Canadian Arctic indicating different aerosol composition and aging.
• Annual mean $eBC$ in 2019, 2021 and 2022 was 33±44, 33±85, and 32±48 ng m-3, respectively.
Mean $eBC$ in the cold and warm periods were equal to 44±47 and 19±57 ng m-3, respectively.
Record high $eBC$ was found in February 2022 (110±70 ng m-3) and August 2021 (83±249 ng
m-3) during the years of study.

The relationship between the magnitude of aethalometer-measured $eBC$ and thermo-optical
EC was evaluated at IBS with respect to the specific atmospheric conditions. $eBC$ mass recalculated
with site-specific absorption coefficient values indicates the observed seasonal effects and gives an
indication of the chemical composition of aerosols. The observed annual cycles show typical Arctic
trends with $eBC$ concentrations higher in the cold period when transport of air masses to IBS occurs
mostly from the European part of Russia, Siberia, Far East, Europe and Asia. When air masses pass
through the oil and gas facilities of Kazakhstan, Volga-Ural, Komi, Nenets, and Western Siberia, BC
contribution from gas flaring dominates over domestic, industrial, and traffic sectors, with a
maximum contribution in January. From June to August, BB source contribution from Siberian fires
exceeds the anthropogenic one. August 2021 and February 2022 showed significant differences with
respect to annual means. During these months, record high BC pollution levels were observed.
August 2021 experienced the worst fire season over the last four decades bringing smoke at IBS.
Vertical distribution of BC at IBS shows a persistent low level BC layer in the cold period, and a
high altitude wildfire smoke layer in the warm period.

Our modelling analyses demonstrate the transport mechanisms from different regions:

• Around 77% contribution from the European part of Russia, Siberia, and Far East was estimated
during the entire study period, followed by Europe (11%), Asia (7%) and North America (4%).
• From December to February, airmass transport from Eurasia (territories above 40ºN), from 1-3
days far from IBS took place, whereas in summer the warm continental airmasses travelled 6-9
days until they reached the receptor.
• The low injection altitude of anthropogenic emissions in cold period leads to a persistent vertical
BC layer at altitudes up to 2 km, with a record high concentration of 100 ng m-3 up to 4 km in



February 2020. In warm period the low BC layer is further elevated due to smoke arriving from the numerous Siberian wildfires.

- Anthropogenic sources contribute 83% of the total for the entire study period; FLR, DOM, TRA, and IND sectors dominate during the Arctic Haze (60%, 22%, 12%, and 9%, respectively). Highest gas flaring (FLR) contribution of 50% and 32% is persistent both in the cold and warm seasons due to the IBS location north of the largest oil and gas producing regions of Western Siberia.

- DOM showed the largest contribution (18%) during the heating season, in agreement with the enhancement of BrC light absorption due to high wood burning contribution from residential emissions.

- During February 2022, modelled BC as 310 ng m-3 when air masses arrived through Western Europe, European part of Russia, and Siberia, passing through the flaring facilities of Kazakhstan, Volga-Ural, Komi, Nenets, and Western Siberia. The overestimated modelled concentrations are likely a result of miscalculated source intensities during these months.

- Annual mean BB contribution approached 48% in the warm season, increasing from April and approaching a maximum (80%) in August. From middle June to September, the monthly mean BB contribution was larger than anthropogenic sources. A few extreme events of high vertical BC at altitudes up to 8 km and 10 km occurred in July 2020 and August 2021, respectively, because of the ongoing wildfires.

- In 2021, the monthly mean spring BB contribution approached a maximum of 36% (May) due to the strong agricultural fires in Siberia. In August 2021, the IBS received an extremely high BB contribution of 90% from the wildfires in Yakutia located 2000 km away.

The increasing trends of occurrence, intensity and duration of wildfires, especially at high northern latitudes, greatly reinforce the importance of carbonaceous light absorption measurements in these areas. These measurements offer valuable insights into the radiative properties of Arctic aerosols, which play a significant role in the enhanced radiative forcing, particularly at short UV-VIS wavelengths. This increased radiative forcing can lead to a pronounced warming effect in the Arctic, with the impact being especially strong during the summer months.

**Data availability.** All modelling data from this study are available for download from https://atmo-access.nilu.no/BELY2_MSU.py. FLEXPART version 10.4 model can be downloaded from https://www.flexpart.eu/downloads. Black Carbon observations are available upon request from O. B. Popovicheva.

**Supplement.** The supplement related to this article in available online at.



666

**Author contributions.** OBP supervised the station operation, interpreted data and wrote the manuscript. NE performed all the FLEXPART simulations and analyses, wrote and coordinated the paper. *MAC* analysed the data, prepared the figures and assisted in the interpretation of the results. ED provided supported *AAE* calculations and evaluation of data quality. NSK supported the research. All authors contributed to the final version of the manuscript.

**Competing interests.** The authors declare no competing interests.

**Acknowledgements.** This research was performed in the frame of the development program of the Interdisciplinary Scientific and Educational School of M. V. Lomonosov Moscow State University "Future Planet and Global Environmental Change". Authors thank Magee Scientific for AE33 instrumentation support and Dr. Asta Gregorič for data examination. V.O. Kobelev is acknowledged data analyses over all study years.

**Financial support.** The article processing charges for this publication were paid by NILU. Development of the methodology for aethalometric measurements and data treatment was performed in the frame of the RSF project #22-17-00102. Institute of Enviromental Survely, Planing and Assessment (IESPA) partly supported the instrumentation and power supply of IBS.

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



**FIGURES & LEGENDS**

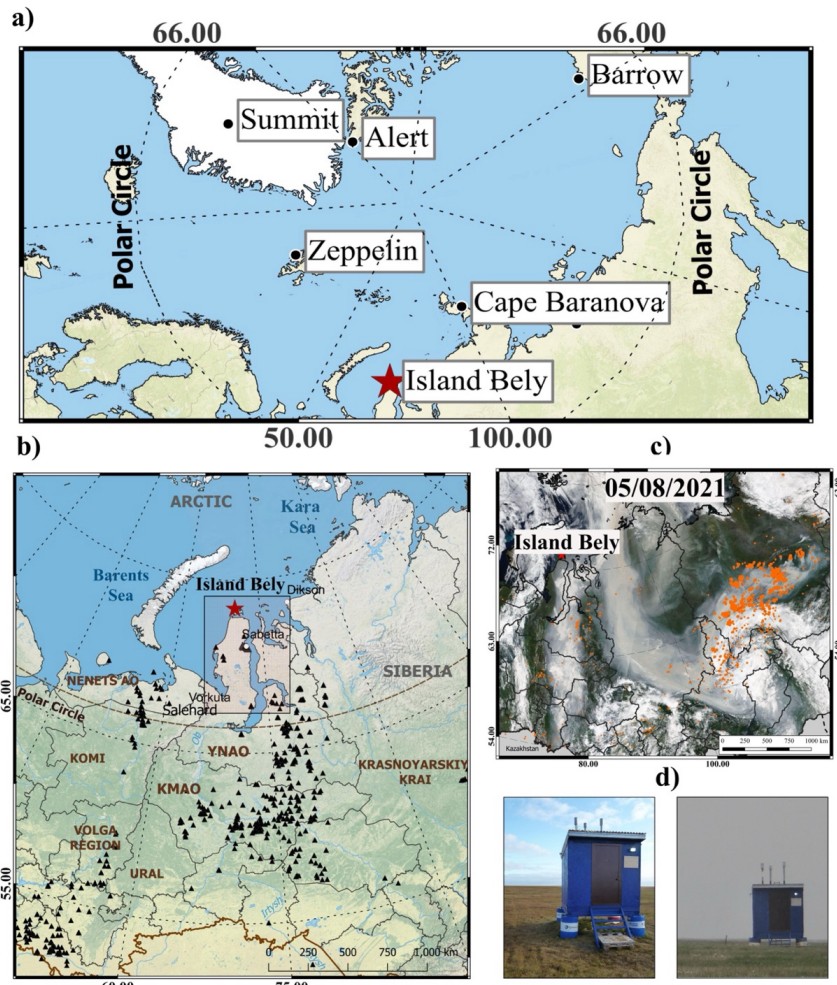

**Figure 1.** (a) The "Island Bely" station (IBS) between other polar aerosol stations. (b) A map
showing IBS in Western Siberia along with oil and gas fields (adopted from https://skytruth.org/,
last access: 7 November 2024). Flares of oil and gas fields are indicated for 2020 as black triangles
(https://skytruth.org/, last access: 7 November 2024). (c) Satellite image of strong plume from the
area of Yakutian wildfires which brought deep smoke to the Bely Island. (d) View to the pavilion of
IBS under clear conditions on 25 July 2021, and during the unprecedented smoke event on 5 August
2021. Maps were created using Open-Source Geographic Information System QGIS
(https://qgis.org/en/site, last access: 7 November 2024) with ESRI physical imagery
(https://server.arcgisonline.com/ArcGIS/rest/services/World_Physical_Map/MapServer/tile/%7Bz%
7D/%7By%7D/%7Bx%7D&zmax=20&zmin=0, last access: 7 November 2024) as the base layer,
and for MODIS Reflectance true color imagery (MODIS Science Team) and Satellite imagery from
05 of August 2021 (https://worldview.earthdata.nasa.gov, last access: 7 November 2024) with
TERRA MODIS fire anomaly layer. Open-source Natural Earth quick start (NEQS) package was
used to add layers of natural and cultural boundaries and polygons from ESRI Shapefile storage.



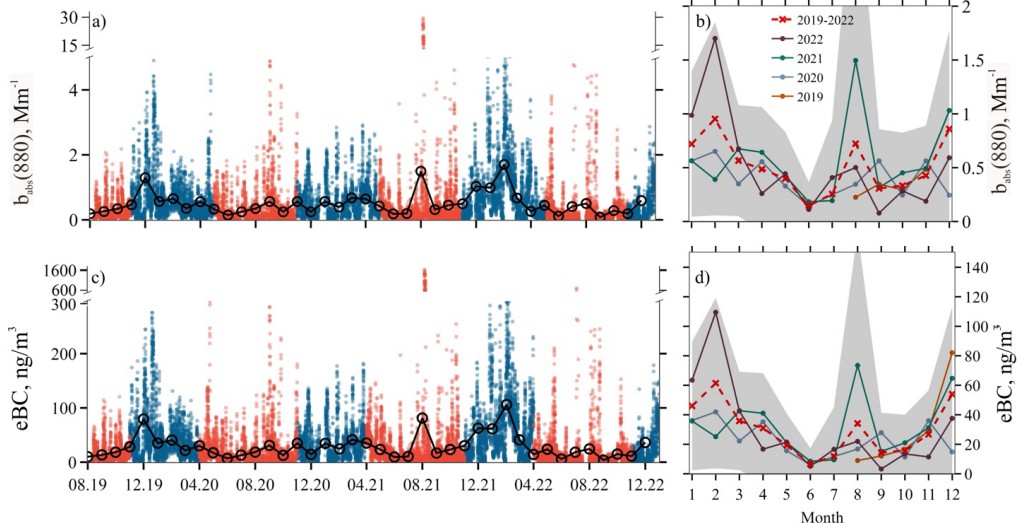

1075

**Figure 2.** Hourly timeseries and monthly means of (a) $b_{abs}(880)$ and (c) $eBC$ for cold (blue) and
warm (red) periods; monthly climatology of (b) $b_{abs}(880)$ and (d) $eBC$ for half year 2019 and 2020,
2021, and 2022. Cross-marks (x) joined by lines show the inter-annual mean; the standard deviation
is plotted by shadow area.

1080

1081



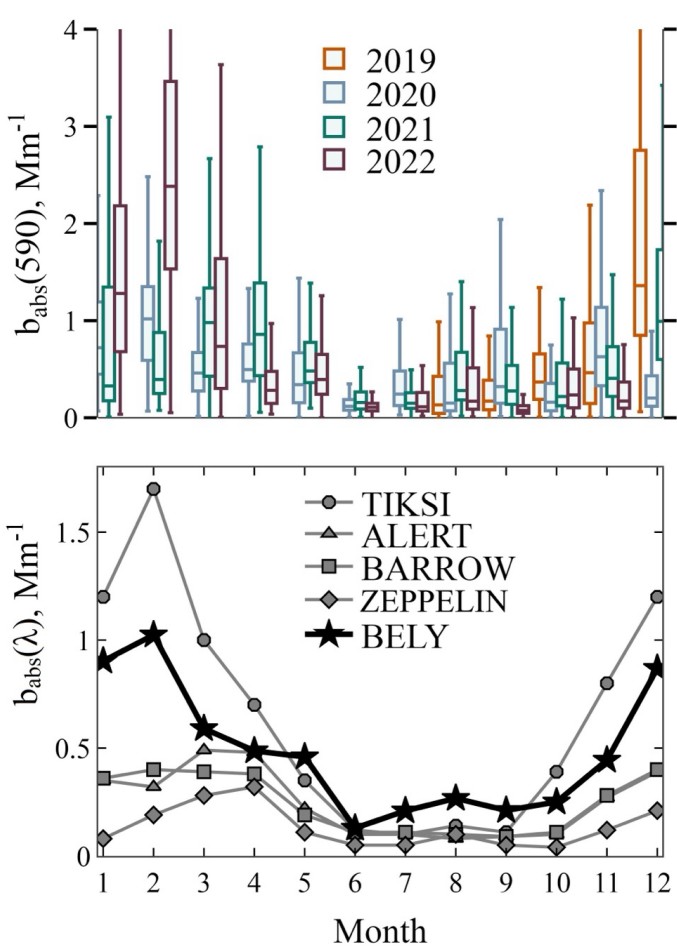

**Figure 3. (a)** Monthly box-whisker plot for $b_{abs}(550)$ at IBS for half year 2019 and full 2020, 2021, and 2022. The 25th, 50th, and 75th percentiles are shown with boxes, while whiskers extend ±1.5 times the interquartile range. (b) Seasonality of monthly median of $b_{abs}$ at 550 nm at Tiksi, Alert, Barrow, Zepelin for 2012-2014 (Schmeisser et al., 2018), and $b_{abs}$ at 590 nm at IBS for 2019-2022 (this work).



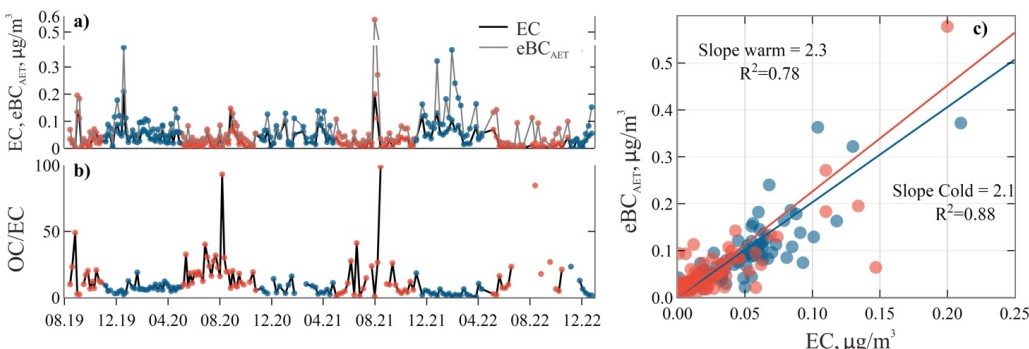

**Figure 4.** Temporal variation of (a) weekly EC and $eBC_{AET}$ averaged over the whole sampling period and (b) the OC/EC ratio. (c) Scatter plots and orthogonal regressions (solid lines) for measured $eBC_{AET}$ and EC concentrations in cold (blue) and warm (red) period. The figure includes the regression slope, the coefficient of determination (R2).



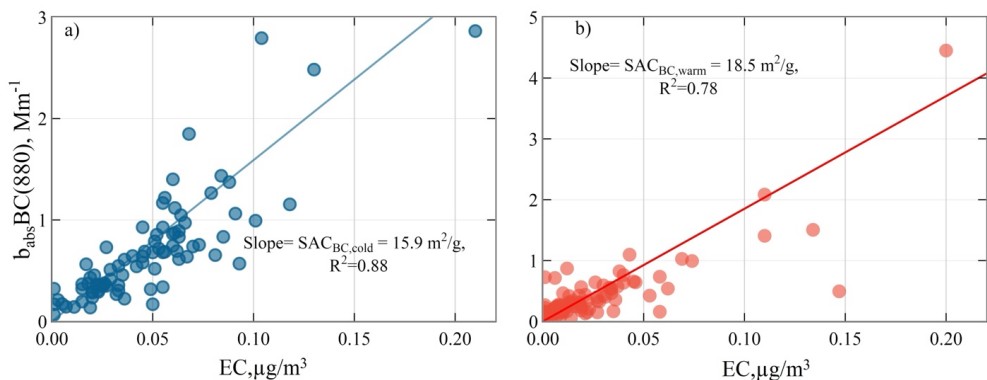

**Figure 5.** Scatter plots and orthogonal regressions (solid line) for $b_{abs/BC}(880)$ and EC concentrations for the (a) cold (blue) and (b) warm (red) periods. Regression slope defines $SAC_{BC,cold}$ and $SAC_{BC,warm}$.



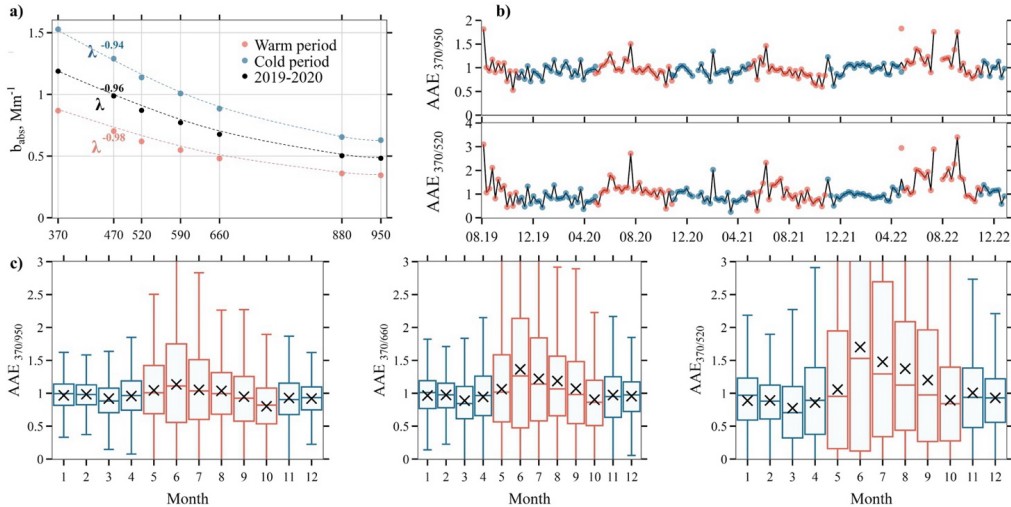

1102

**Figure 6. (**a) Spectral dependence of light absorption coefficient for 2019-2022, during warm (red) and cold (blue) periods. $AAE_{350/950}$ is the slope of the linear regression in logarithmic scale of a power law regression, Eq.1. (b) Timeseries of $AAE_{370/950}$ and $AAE_{370/520}$. (c) Box-whisker plots and monthly means of $AAE$ at 370 and 950 nm, 370 and 660 nm, and 370 and 520 nm for the entire period.

1108



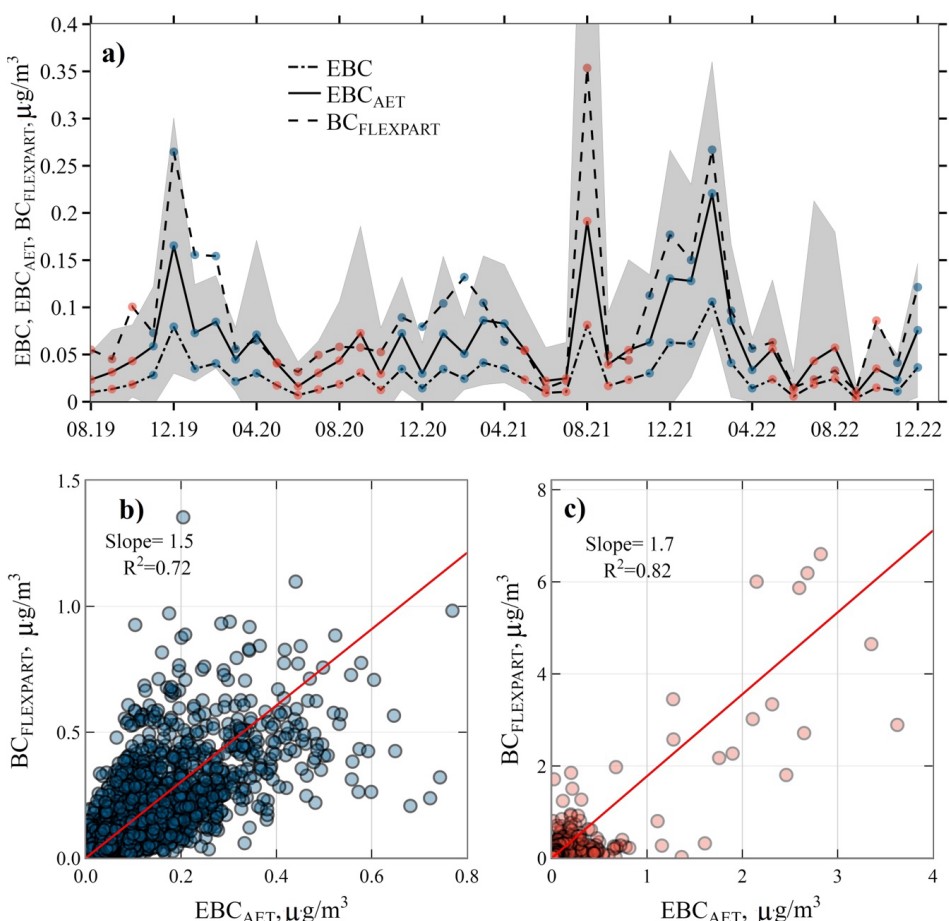

**Figure 7.** (a) Monthly mean $eBC$ and modelled surface BC concentrations from 10 August 2019 to 31 December 2022. Monthly mean $eBC_{AET}$ (line with crosses) shown with the standard deviation range by shadowed area. Scatter plots and orthogonal regressions (solid lines) for $BC_{FLEXPART}$ calculated over measured $eBC_{AET}$ concentrations for (b) cold and (c) warm period. The figure includes the regression slope, the coefficient of determination (R2).



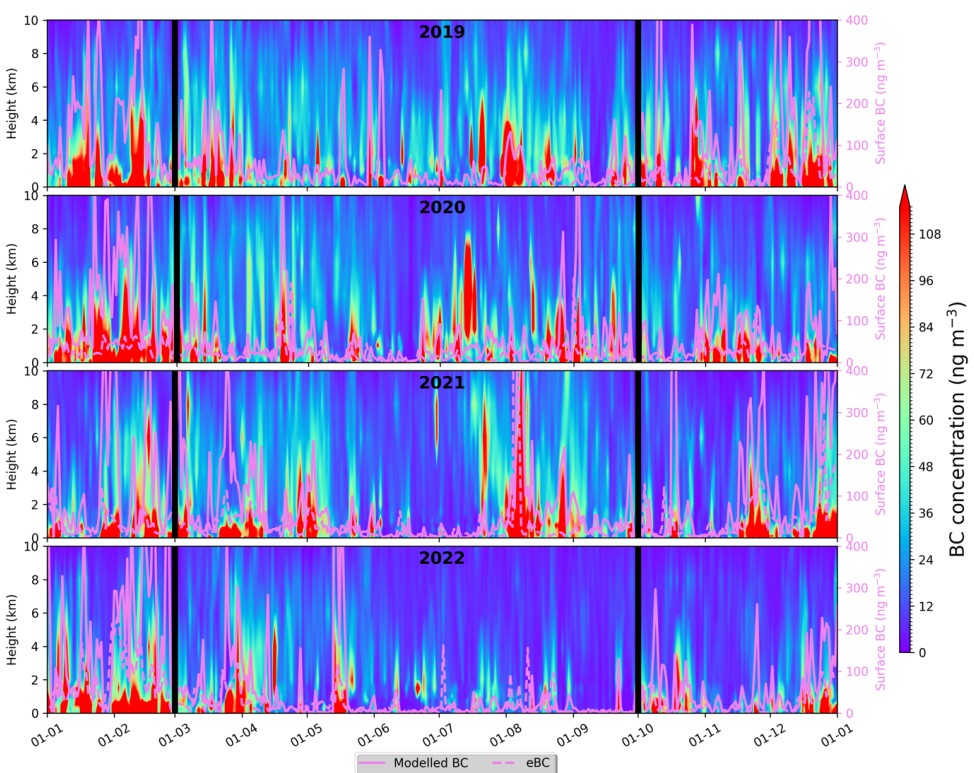

**Figure 8.** Vertical cross-sections of modelled BC for 2019-2022. Solid and dotted violet lines represent modelled daily surface BC and $eBC$, respectively. Their levels correspond to the right (secondary) axis (also in violet). Boundaries between the cold (November- April) and warm (May-October) are indicated by thick vertical black lines.



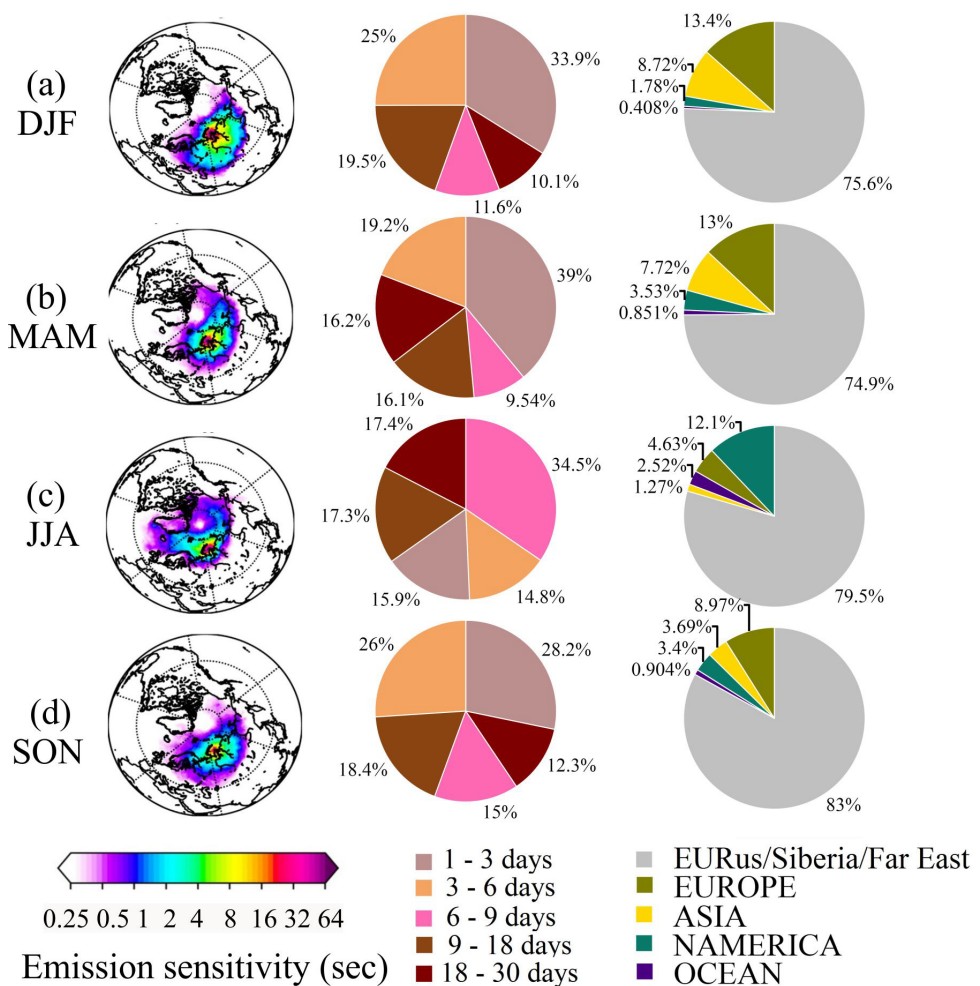

**Figure 9.** (a-d) Season average footprint emission sensitivity, mean age contribution of emissions from different day-periods back in time and each region contribution to surface concentration of BC.

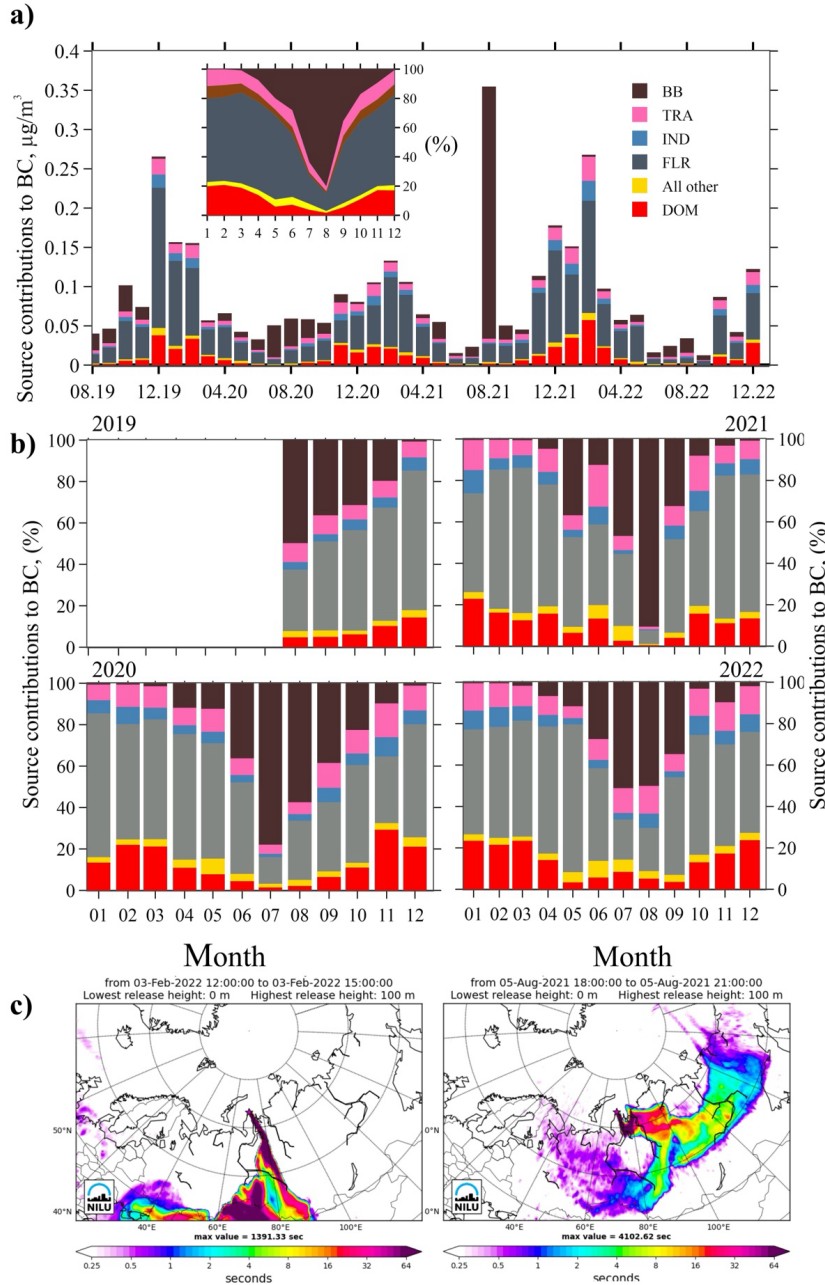

**Figure 10. (**a) Timeseries of monthly mean contribution from different emission source types to
surface BC concentrations for the study period. (b) Percentage of monthly mean source
contributions for each year. Residential and commercial (DOM), biomass burning (BB),
transportation (TRA), industrial combustion and processing (IND), gas flaring (FLR), and All
others sources were adopted from ECLIPSEv6 and CAMS GFAS. (c) FES for 3 February 2022 and
5 August 2021 showing the largest probability of emission origin.




**TABLES & LEGENDS**

**Table 1**. Statistics of light - absorption coefficients; EC, OC, $eBC_{AET}$, and $eBC$ mass concentration; absorption Angstrom exponents ($AAE$) for the study period, cold and warm periods. Mean± standard deviation (1σ), 1st and 3rd Q quartile (25th and 75th percentiles).

| Variable | August 2019 - December 2022 | | | | cold (November-April) | | | | warm (May-October) | | | |
|---|---|---|---|---|---|---|---|---|---|---|---|---|
| | mean±sd | median | 1st Q | 3rd Q | mean±sd | median | 1st Q | 3rd Q | mean±sd | median | 1st Q | 3rd Q |
| $b_{abs}(880)$ (Mm⁻¹) | 0.5±0.9 | 0.3 | 0.1 | 0.6 | 0.7±0.7 | 0.5 | 0.22 | 0.9 | 0.4±0.9 | 0.2 | 0.09 | 0,4 |
| $b_{abs}(520)$ (Mm⁻¹) | 0.9±1.6 | 0.4 | 0.2 | 1 | 1.2±1.2 | 0.8 | 0.38 | 1.5 | 0.6±1.8 | 0.3 | 0.1 | 0,6 |
| $b_{abs}(370)$ (Mm⁻¹) | 1.2±2.4 | 0.6 | 0.3 | 1.4 | 1.6±1.6 | 1.1 | 0.52 | 2.1 | 0.9±2.8 | 0.4 | 0.2 | 0,9 |
| EC (μg m⁻³) | 0.03±0.03 | 0.02 | 0.01 | 0.05 | 0.05±0.03 | 0.04 | 0.02 | 0.06 | 0.02±0.03 | 0.02 | 0.01 | 0,03 |
| OC (μg C m⁻³) | 0.45±0.3 | 0.4 | 0.3 | 0.5 | 0.4±0.2 | 0.4 | 0.3 | 0.5 | 0.5±0.4 | 0.4 | 0.3 | 0,6 |
| * $eBC_{AET}$ (ng m⁻³) | 65±83 | 36 | 16 | 80 | 84±90 | 57 | 25 | 115 | 53±158 | 23 | 10 | 45 |
| * $eBC$ (ng m⁻³) | 29±54 | 13 | 5 | 34 | 44±47 | 29 | 12 | 59 | 19±57 | 8.0 | 4 | 17 |
| $AAE_{370/950}$ | 0.96±0.6 | 0.95 | 0.7 | 1.19 | 0.94±0.4 | 0.95 | 0.74 | 1.1 | 0.98±0.8 | 0.95 | 0.6 | 1,3 |
| $AAE_{370/520}$ | 1.0±1.5 | 0.93 | 0.4 | 1.52 | 0.88±1 | 0.89 | 0.49 | 1.2 | 1.16±1.9 | 1.0 | 0.3 | 2,0 |

* $eBC_{AET}$ is defined in section 2.2.
** $eBC$ is defined in section 3.2.