# Peer review of "Multi-year black carbon observations and modeling close to the"

_EGUsphere, 2024_

## Referee Comment (RC1)

**Review**

The manuscript entitled "Multi-year black carbon observations and modeling close to the largest gas flaring and wildfire regions (Western Siberian Arctic)" by Popovicheka et al. presents a detailed study on Black Carbon (BC) measurments in a new Arctic station in Kara Sea. The study investigates the seasonal differences in measured optical properties and assess the inter-annual variability of BC sources using an atmpsheric dispesion model. The manuscript is well structured for the readers to understand the analytical steps and the results obtained.

I recommend the publication of the manuscript following minor revisions.

First I would like to mention a few general comments which apply to the whole manuscript:

(1) Please define the abbreviations the first you are using them and not each time in a new sentence, section or Table.

(2) Be careful with the units. For BC/eBC you either use ng m$^{-3}$ or μg m$^{-3}$, and then you present your statistics in ng m$^{-3}$ . Please choose one and do not show your results in both. It's confusing. Also be consistent and use ng m$^{-3}$ and not ng/m3. Correct everywhere.

(3) The numbers in the units must to be in superscript. Correct everywhere.

Minor comments (The number of the lines below correspond to the submitted manuscript):

Lines 15-31: I am missing one-two sentences on FLEXPART and which are the main emissions sources the authors found to be contributing to BC at the station.

Line 24: Define eBC

Lines 44-45: Please consider rephrasing this sentence – it's to vague. Which are the complicated processes? Define Arctic haze.

Lines 52-53: What about the latest AMAP report? Does the latest report provide updated results on forcing? If yes, please updated accordingly.

Lines 62-63: "The most complicated issue .." Such as? Rephrease or be more specific.

Lines 70-72: If you link these sentences with your text earlier please rephrase - otherwise say why, it is not clear.

Line 82 and wherever mentioned again: "Barrow". Please consider using the local name Utqiagvik or as Barrow/Utqiagvik to be more respectul to the local community.

Line 88: What about the resutls presented Matsui et al. 2022?

Line 89: replace "in" with "is"

Lines 105-106: Too vague. Please say a bit more here.

Lines 119-120: add a reference for your statement: "as the station is located along the main pathway of large-scale emission plumes from industrial regions and Siberian wildfires entering the Arctic."

Line 128: AAE – in line 73 – no need to define again

Line 132: Add a sentence on what the following sections show.

Line 143: ".. snow coverage, wind speed, and relative humidity ... "

Lines 145-146: On what criteria you base the split in two periods? If possible, mention.

Lines 144-149: Instead of providing an annual value please provide a value for the two periods (cold/warm) – would be easier for the reader to get an overall idea – as you do in lines 150-154

Lines 155-161: Move this part in the beginning of the following sub-section (2.2) – it fits better there.

Line 179 – equation 2: Define $b_{abs}$ in equation 1

Line 213: "Both methods have important uncertainties ..": Please give percentanges/numbers. Also discuss/mention AE33 uncertainties here.

Line 235: Consider rephrasing the title to "Atmosphere disperion modelling and emissions" or something similar

Line 235: From the text you provide here it is not clear for which periods/years you run the model. Please mention.

Line 241: Are 30 days sufficient period to account for Arctic Haze? Did you test longer periods? Where do you base the 30 days? Please justufy your choice if possible.

Line 243: What is tracking?

Lines 269-272: Mention that the satellite image is shown in Figure 1e. Also is there any particular reason you are mentioning this here? You could simpply mention it once directly when you discuss Figure 1e.

Line 272 and afterwards: Please consider discussing the main patterns for temperature, wind speed and direction in Section 3. It would be more interesting to see there the wind patterns.

Lines 280-284: Do you know why? If yes, please discuss.

Lines 329-331: Why? Please clarify

Lines 345-358: You compare your results with other studies at Zeppelin. What about at other Arctic sites where there published papers even if the period is a bit shorter, e.g. Barrow/ Utqiagvik?

Line 372: Why are you showing eBC in Fig. 2? You start discussing about eBC/BC from Fig. 4. Consider showing eBC time series later.

Line 376: add ", respectively" after Table S2.

Line 377: You mention p-value here for eBC. Could you calculate the p-value for all the measurments you provide? Also mention how did you calculate the p-value.

Line 413: It seems it's missing something before the parenthesis "(370 and 520 nm)"

Line 444: Do you mean 0.82 instead of 0.7? Based on the Figure 7.

Line 445: What about biases? Can you calculated them as well and provide the numbers?

Line 452: Do you think if you have used CAMS emissions, with a resolution 10 km$^2$, you would have got better results?

Lines 461: How many vertical levels did you use in the model? Did you allocate the emissions in the surface or did you distribute them at different levels? Please mention in the relevant section.

Line 574 – Conclusions: This section is way too long. It also reads as a grocery list and not as a scientific discussion. Please consider re-writting this section by interpretating you main findings and the main take away message the reader should take.

Line 665: The link for the supplement material is missing. Please provide.

Lines 1059-1074: Please consider correcting Barrow to Barrow/Utqiagvik. In the manuscript you mention Tiksi, Pallas and Villum. Please show their location in the Figure.

Lines 1082-1088: Figure 3.  Please consider correcting Barrow to Barrow/Utqiagvik.

Lines 1075-1081: Figure 4. Add a legend in Fig.3a showing cold and warm period.  In the b$_{abs}$ time series there is a peak after 04.20 showing in blue. For eBC the same peak is shown in red. Why? The same goes for 04.21. Also please consider changing the format of the date axis in this figure and whenever else (in the rest of the figures) is applicable. It took me some time to realise these are dates. The format you are using is confusing.

Lines 1102-1108: Figure 6 - In figure 6a you mention "-0.96" while in the text you discuss 0.96. Why? Is 2019-2020 included in the warm/cold period? If no, why?

Lines 1109-1115: Figure 7  - What do the slopes show? Please mention

Lines 1116-1121: Figure 8 – In the title you refer to the station as "Bely station". Please be consistent and refer to the station as "Island Bely" station

---

## Referee Comment (RC2)

Reviewer ID# 2
Manuscript No.: egusphere-2024-3124
Authors: Olga B. Popovicheva et al.
The title of manuscript: **Multi-year black carbon observations and modeling close to the largest gas flaring and wildfire regions (Western Siberian Aretic)**

**General comments:**

The Arctic has warmed three times more quickly than the planet as a whole, making it the most sensitive region to climate change. To understand the impacts of BC emissions on the Arctic from source regions, particularly from the Siberian Arctic, the authors present three and a half years measurements of equivalent BC (eBC) concentrations from 2019 to 2022. These measurements are complemented by elemental carbon (EC) data obtained using therma-optical method, conducted at the recently established station "Island Bely" (IBS). The station is located along the main pathway through which the highest levels of anthropogenic pollution from industrial regions, as well as emissions from Siberian wildfires, enter the Arctic.

Furthermore, the authors evaluate seasonal variations in intensive optical properties and their dependence on wavelength, as indicated by Absorption Angstrom Exponent (AAE). They also estimate the site-specific absorption coefficient (*SAC*), which serves as the basis for calculating eBC values.

By coupling the FLEXPART Lagrangian particle dispersion model with the latest black carbon (BC) emission inventories for anthropogenic and biomass burning sources, the study investigates the detailed aerosol aging spectrum, source region attribution, and source sector apportionment for the entire period as well as during pollution episodes. This approach effectively integrates observations with model simulations and emission inventories.

However, it is noted that this work appears very similar to a study published in 2022 by the same group (Popovicheva et al., 2022), conducted at the same location with similar observations and using the same model. The conclusions are largely the same, apart from the inclusion of two additional years of observations.

I would recommend publishing the manuscript in *Atmospheric Chemistry and Physics* (ACP) after the authors address the following aspects:

- Reduce or condense repetitive content that overlaps with the previous paper published in 2022.

- Emphasize aspects that are being discussed for the first time in this study, such as "BrC and its relationship with the corresponding AAE" and "eBC (derived from SAC) and its comparison with eBCAET".

- It is noted that the eBC values are approximately half of the $eBC_{AET}$ values (Table 1). Could the authors have a further discussion regarding the difference?

- There appears to be a significant discrepancy between the results in Table S3 and Figure 10 b). For example, in July 2020, Figure 10 b) shows that biomass burning (BB) accounts for about 80%, whereas Table S3 indicates only a 7% contribution of BrC during the same period. Conversely, in February 2020, Figure 10 b) shows less than 10% BB, while Table S3 also reports a 7% BrC contribution. Could the authors clarify why these differences occur?

**Specific comments:**

L27-L28: Is the "92%" an average value of cold seasons over the entire study period?

It appears that the 92% mentioned in abstract is not consistent with the 83% stated in the conclusion.

L87-L88: Are the Biomass Burning (BB) results in Figure 10 a) and b) from ground level or from the altitudes (600-800 hPa)? Do the BB results in Figure 10 support the previously work by Qi and Wang, 2019?

L150: Should this be Fig. S1, instead of Fig.1d ?

L159: What are the two inlets with a much higher flow rate (~ 38LPM) than 5LPM?

L170: Please specify $eBC_{AET}$ as "equivalent black carbon concentration by aethalometer", when mentioning it for the first time.

L182: Please spell out for "NIR-VIS"

L240: What is the highest level of the 137 vertical levels?

How do you couple with the BB burning injection height in the emission inventory?

L276-L277: Please be consistent with the wavelength mentioned (880nm or 800 nm?). Only 880nm is shown on Fig. 2.

L277: The expression is confusing, i.e., "the mean ± sigma (median) values".

 It is suggested to use "the mean ± sigma and the median, respectively".

L285: Please ensure consistency between the text and Table 2.

L295-L297: The sentence is not well expressed. Please consider re-phrasing it.

L308 & L1083: Please ensure consistency in the wavelength between the text on page 11 and Figure 3(a). The caption for Figure 3(a) should indicate 590 nm. Additionally, please add (a) and (b) to Figure 3.

L378: Is it the cold season, or the warm season over the entire period?

L385-L388: The numbers in the text differ slightly from those in Table S2. Please ensure consistency between the text and the table.

L427: No box-whisker plots are shown in Figure S3, which presents the emissions of BC from CAMS-GFAS. This should be Figure S4.

L431: It is suggested to add "in percentage" (shown in Table S3) after the "contribution".

L439: Can the authors explain why $eBC_{AET}$ is higher than eBC by a factor of two?

L498: Should it be Figure 9?

L520-L521: It is interesting to see a lack of consistency between Figure 10b and Table S3. It appears that a high BB contribution does not result in a relatively high percentage of BrC.

L546: It should be ~ 50%, instead of 60%.

L547: It is suggested to use 72,400 $km^2$.

L560: what is the unit of "150,000"?

L574: In general, the 'Conclusion' section is too long and should be more concise.

L596: "BrC light absorption coefficient in the UV spectrum showed similar trends as BC, although it exceeded BC by 2.4 times during both cold and warm periods." What is the basis of this statement?

L607: Please confirm the numbers. They should be consistent with these in Table S2, where the numbers are 106 ± 67 ng $m^{-3}$.

L669: Please use "MAC", instead of "*MAC*" to ensure consistency when writing the initials. Otherwise, it would be confused with the Mass Absorption Coefficient (*MAC*).

---

## Author Response (AR1)

Comments from Reviewer 1

The manuscript entitled "Multi-year black carbon observations and modeling close to the largest gas flaring and wildfire regions (Western Siberian Arctic)" by Popovicheka et al. presents a detailed study on Black Carbon (BC) measurements in a new Arctic station in Kara Sea. The study investigates the seasonal differences in measured optical properties and assess the inter-annual variability of BC sources using an atmospheric dispersion model. The manuscript is well structured for the readers to understand the analytical steps and the results obtained.

I recommend the publication of the manuscript following minor revisions. First I would like to mention a few general comments which apply to the whole manuscript:

(1) Please define the abbreviations the first you are using them and not each time in a new sentence, section or Table.

**Response**: We have defined the abbreviations the first time they are used. Notice that many of them are now defined in the Abstract (see Manuscript with Track Changes). To help to verification, we list all the abbreviations we have used below:

black carbon (BC)

equivalent black carbon $(eBC)$

mass-specific absorption coefficient $(MAC)$

absorption Angstrom exponent $(AAE)$

brown carbon (BrC)

biomass burning (BB)

"Island Bely" Station (IBS)

elemental carbon (EC)

site-specific absorption coefficient $(SAC)$

aethalometer-produced equivalent black carbon concentration $(eBC_{AET})$

thermo-optical transmittance (TOT)

carbonate carbon (CC)

Lagrangian particle dispersion model FLEXPART

industrial combustion (IND)

energy production (ENE)

residential and commercial emissions (DOM)

waste treatment and disposal sector (WST)

transportation (TRA)

shipping (SHP)

gas flaring emissions (FLR)

Evaluating the CLimate and Air Quality ImPacts of ShortlivEd Pollutants (ECLIPSE)

Copernicus Global Fire Assimilated System (CAMS GFAS)

Light-absorption coefficients at 880 nm ($b_{abs}(800)$)

Light absorption at 370 nm ($b_{abs}(370)$)

Modelled surface BC ($BC_{FLEXPART}$)

European part of Russia (EURus)

(2) Be careful with the units. For BC/eBC you either use ng m-3 or µg m-3, and then you present your statistics in ng m-3 . Please choose one and do not show your results in both. It's confusing. Also be consistent and use ng m-3 and not ng/m3. Correct everywhere.

**Response**: We completely agree with this comment. We have changed all units to ng m-3 (see Manuscript with Track Changes).

(3) The numbers in the units must be in superscript. Correct everywhere.

**Response**: We have now tried to put all units in superscript (see Manuscript with Track Changes). From previous experience, this is usually taken over during typesetting (in case the manuscript is accepted), as we might have missed something.

Minor comments (The number of the lines below correspond to the submitted manuscript):

Lines 15-31: I am missing one-two sentences on FLEXPART and which are the main emissions sources the authors found to be contributing to BC at the station.

**Response**: We have added two lines answering to the question about the dominating sources (Lines 28-30, Manuscript with Track Changes)

Line 24: Define eBC

**Response**: eBC has been defined in Line 25 (Manuscript with Track Changes).

Lines 44-45: Please consider rephrasing this sentence – it's too vague. Which are the complicated processes? Define Arctic haze.

**Response**: We have rephrased the sentence and defined Arctic Haze in Lines 49-52 (Manuscript with Track Changes).

Lines 52-53: What about the latest AMAP report? Does the latest report provide updated results on forcing? If yes, please updated accordingly.

**Response**: To our knowledge, the latest AMAP report is AMAP (2021) as given in the manuscript. Please see the AMAP publication website here: https://www.amap.no/publications .

Lines 62-63: "The most complicated issue .." Such as? Rephrease or be more specific.

**Response**: Rephrased (Lines 71-72, Manuscript with Track Changes).

Lines 70-72: If you link these sentences with your text earlier please rephrase - otherwise say why, it is not clear.

**Response**: We have used an adverb to link the two sentences (Lines 86, Manuscript with Track Changes).

Line 82 and wherever mentioned again: "Barrow". Please consider using the local name Utqiagvik or as Barrow/Utqiagvik to be more respectul to the local community.

**Response**: We have added Barrow/Utqiagvik all over the manuscript (Lines 98, 374, 376, 1206, Manuscript with Track Changes).

Line 88: What about the resutls presented Matsui et al. 2022?

**Response**: We added some comments about the results from Matsui et al. (2022) in Lines 104-106 (Manuscript with Track Changes).

Line 89: replace "in" with "is"

**Response**: Changed in Line 107 (Manuscript with Track Changes).

Lines 105-106: Too vague. Please say a bit more here.

**Response**: We have rephrased in Lines 132-134 (Manuscript with Track Changes).

Lines 119-120: add a reference for your statement: "as the station is located along the main pathway of large-scale emission plumes from industrial regions and Siberian wildfires entering the Arctic."

**Response**: We have shown this in Popovicheva et al. (2022), now added in Line 149 (see Manuscript with Track Changes).

Line 128: AAE – in line 73 – no need to define again

**Response**: Corrected (Line 164 in Manuscript with Track Changes).

Line 132: Add a sentence on what the following sections show.

**Response**: We do not consider this is relevant here, since the following sections have distinct titles, and we believe it is not necessary to plagiarize in an already long paper. We are willing to do this at a later stage and if the Editor thinks this is important for the flow of the paper.

Line 143: ".. snow coverage, wind speed, and relative humidity ..."

**Response**: "… and" removed from the sentence (see Line 178 in Manuscript with Track Changes).

Lines 145-146: On what criteria you base the split in two periods? If possible, mention.

**Response**: We have now written that the definition of cold and warm period is based on the prevailing temperatures (see Lines 180-181 in Manuscript with Track Changes).

Lines 144-149: Instead of providing an annual value please provide a value for the two periods (cold/warm) – would be easier for the reader to get an overall idea – as you do in lines 150-154

**Response**: Corrected, see Lines 277-278 (Manuscript with Track Changes).

Lines 155-161: Move this part in the beginning of the following sub-section (2.2) – it fits better there.

**Response**: Moved as the first paragraph of section 2.2 (see Line 193 in Manuscript with Track Changes).

Line 179 – equation 2: Define babs in equation 1

**Response**: Corrected (see Line 193 in Manuscript with Track Changes).

Line 213: "Both methods have important uncertainties ..": Please give percentanges/numbers. Also discuss/mention AE33 uncertainties here.

**Response**: We reported uncertainties in Line 303 (see Manuscript with Track Changes). We have not discussed uncertainties of BC instruments, since this paper in not a measurement or methodological review. We point to Petzold et al. (2013), Sharma et al., 2017 and Ohata et al., 2021) (see references therein Manuscript).

Line 235: Consider rephrasing the title to "Atmosphere disperion modelling and emissions" or something similar

**Response**: We have changed the title in section 2.3 to "Atmospheric dispersion modelling and emission inventories".

Line 235: From the text you provide here it is not clear for which periods/years you run the model. Please mention.

**Response**: Corrected (see Line 307 references therein Manuscript).

Line 241: Are 30 days sufficient period to account for Arctic Haze? Did you test longer periods? Where do you base the 30 days? Please justify your choice if possible.

**Response**: 30 days backward in time is sufficient to capture the majority of the emissions arriving at the receptor given a lifetime of 5-7 days for BC. In fact, the fraction of the emissions arriving at the receptor for 30 days tracking has been calculated to be >98%. Please see baseline papers such as Stohl et al. (1998; 2005) and Pisso et al. (2019) (references therein Manuscript).

Line 243: What is tracking?

**Response**: "Tracking" in a common term in Lagrangian modelling corresponding to the computational particle tracking. It is explained 2 lines above "… computational particles … and tracked backward …". Please see baseline paper of Seibert and Frank (2004): https://doi.org/10.5194/acp-4-51-2004.

Lines 269-272: Mention that the satellite image is shown in Figure 1e. Also is there any particular reason you are mentioning this here? You could simply mention it once directly when you discuss Figure 1e.

**Response**: This is correct, and we appreciate for the suggested correction. We have now moved the lines where Figure 1 is mentioned for the first time (around Line 177 in Manuscript with Track Changes).

Line 272 and afterwards: Please consider discussing the main patterns for temperature, wind speed and direction in Section 3. It would be more interesting to see there the wind patterns.

**Response**: The description of wind patterns, precipitation etc… fall within the station characteristics and they are not actual results of our study. They are not discussed further in the Results section and therefore we intend to let them in section 2.1 in order to not disturb the sequence of our manuscript.

Lines 280-284: Do you know why? If yes, please discuss.

**Response**: It is well known that the seasonal variation resulting in the Arctic Haze in winter is caused by the formation of the Polar Dome. In our opinion, this is rather trivial to be mentioned in such a long paper, but we now explain it briefly (see Line 577 in Manuscript with Track Changes).

Lines 329-331: Why? Please clarify

**Response**: We have added a brief explanation (Lines 534-537 in Manuscript with Track Changes).

Lines 345-358: You compare your results with other studies at Zeppelin. What about at other Arctic sites where there published papers even if the period is a bit shorter, e.g. Barrow/Utqiagvik?

**Response**: Other Arctic stations such as Barrow experience completely different source contributions. In addition, in this section, we only aim to present the results during our intensive campaign in IBS. However, we complement the results from IBS with those from most of the other Arctic station in Figure 3.

Line 372: Why are you showing eBC in Fig. 2? You start discussing about eBC/BC from Fig.4. Consider showing eBC time series later.

**Response**: The section, where Line 372 is places, is entitled "3.2 Black carbon and site-specific mass absorption cross-section". Figure 2 that corresponds to this section shows exactly what is mentioned on the title, namely babs(880) on top and eBC at the bottom. We do not see why this needs to be changed.

Line 376: add ", respectively" after Table S2.

**Response**: Added in Line 369 (Manuscript with Track Changes).

Line 377: You mention p-value here for eBC. Could you calculate the p-value for all the measurements you provide? Also mention how did you calculate the p-value.

**Response**: We do not understand how bombarding the readers with p-values would ease the flow of the manuscript. All the calculated p-values were < 0.05. The calculation is easy to be done with statistical software and even easier with object-oriented tools such as python. Accordingly, since we have a t-test for independent samples the t statistic is:

$$t = \frac{\bar{X}_1 - \bar{X}_2}{\sqrt{\frac{s_1^2}{n_1} + \frac{s_2^2}{n_2}}}$$ where $\bar{X}_1, \bar{X}_2$ the sample means, $s_1$ , $s_2$ the variances and $n_1, n_2$ the size of the samples.

The p-value is obtained from the t-distribution:

p=2×P(T>│t│)

where P(T>│t│) is the probability of observing a t-value greater than the computed one, based on the t-distribution with df degrees of freedom (computed with Welch's formula).

Line 413: It seems it's missing something before the parenthesis "(370 and 520 nm)"

**Response**: Corrected (see Line 518, Manuscript with Track Changes).

Line 444: Do you mean 0.82 instead of 0.7? Based on the Figure 7.

**Response**: We appreciate for pointing this typo. Now corrected in Line 554.

Line 445: What about biases? Can you calculate them as well and provide the numbers?

**Response**: We added calculated bias in Lines 565-566 (see Manuscript with Track Changes).

Line 452: Do you think if you have used CAMS emissions, with a resolution 10 km2, you would have got better results?

**Response**: The reason why we use 0.5 degree resolution emission inventories is because the meteorological fields we use in FLEXPART have the same resolution. If we go higher resolution using 0.5 degrees resolution in the meteorological input (ECMWF ERA5), it is simply an interpolation. Of course, using high-resolution emissions is always an asset, but it is computationally inefficient when focusing on long-range transport mechanisms.

Lines 461: How many vertical levels did you use in the model? Did you allocate the emissions in the surface or did you distribute them at different levels? Please mention in the relevant section.

**Response**: The model uses 137 vertical levels extending from the surface until approximately 80 km. The anthropogenic releases occur at ground level (0-100 m). The BB emissions occur at higher altitudes. We have mentioned this in Lines 345-347 of the Methods section.

Line 574 – Conclusions: This section is way too long. It also reads as a grocery list and not as a scientific discussion. Please consider re-writing this section by interpretating you main findings and the main take away message the reader should take.

**Response**: The Conclusions section has been re-written and shortened as reviewer suggested. Please see Manuscript with Track Changes.

Line 665: The link for the supplement material is missing. Please provide.

**Response**: The link to the supplements is added by the journal if the manuscript is published. The Supplements for review are provided as an external file, at present.

Lines 1059-1074: Please consider correcting Barrow to Barrow/Utqiagvik. In the manuscript you mention Tiksi, Pallas and Villum. Please show their location in the Figure.

**Response**: We have corrected this everywhere in the manuscript (see previous comments and Lines 401, 403, 1253 (Manuscript with Track Changes). We also show the location of the other Arctic stations in Figure 1.

Lines 1082-1088: Figure 3. Please consider correcting Barrow to Barrow/Utqiagvik.

**Response**: Please see line 1253 (legend of Figure 3). We have not added local name Utqiagvik in the Figures, because the name becomes too long and the figures overloaded.

Lines 1075-1081: Figure 4. Add a legend in Fig.3a showing cold and warm period. In the babs time series there is a peak after 04.20 showing in blue. For eBC the same peak is shown in red. Why? The same goes for 04.21. Also please consider changing the format of the date axis in this figure and whenever else (in the rest of the figures) is applicable. It took me some time to realise these are dates. The format you are using is confusing.

**Response**: We struggle to understand what the reviewer means here. The lines and symbols differ in color to show the cold (blue) and warm period (red). All time axes have been corrected as suggested by the reviewer (see Manuscript with Track Changes).

Lines 1102-1108: Figure 6 - In figure 6a you mention "-0.96" while in the text you discuss 0.96. Why? Is 2019-2020 included in the warm/cold period? If no, why?

**Response**: Equation 2 denotes: $b_{abs}(\lambda) = b_{abs}(\lambda_o) \times \left(\frac{\lambda}{\lambda_0}\right)^{-AAE}$. To have the curve as in Figure 6a, one can easily understand that the power needs to be a positive number.

Lines 1109-1115: Figure 7 - What do the slopes show? Please mention

**Response**: In a scatter plot, the slope of the regression line, m, is given by y = mx + b. So, if the regression line is y=5x+10, then m=5 means that for every 1-unit increase in x, y increased by 5.

Lines 1116-1121: Figure 8 – In the title you refer to the station as "Bely station". Please be consistent and refer to the station as "Island Bely" station

**Response**: Corrected! Please see Figure 8 in the Manuscript with Track Changes.

Comments from Reviewer 2
Manuscript No.: egusphere-2024-3124

Authors: Olga B. Popovicheva et al.

The title of manuscript: Multi-year black carbon observations and modeling close to the largest gas flaring and wildfire regions (Western Siberian Arctic)

General comments:

The Arctic has warmed three times more quickly than the planet as a whole, making it the most sensitive region to climate change. To understand the impacts of BC emissions on the Arctic from source regions, particularly from the Siberian Arctic, the authors present three and a half years measurements of equivalent BC (eBC) concentrations from 2019 to 2022. These measurements are complemented by elemental carbon (EC) data obtained using thermal-optical method, conducted at the recently established station "Island Bely" (IBS). The station is located along the main pathway through which the highest levels of anthropogenic pollution from industrial regions, as well as emissions from Siberian wildfires, enter the Arctic.
Furthermore, the authors evaluate seasonal variations in intensive optical properties and their dependence on wavelength, as indicated by Absorption Angstrom Exponent (AAE). They also estimate the site-specific absorption coefficient (SAC), which serves as the basis for calculating eBC values.
By coupling the FLEXPART Lagrangian particle dispersion model with the latest black carbon (BC) emission inventories for anthropogenic and biomass burning sources, the study investigates the detailed aerosol aging spectrum, source region attribution, and source sector apportionment for the entire period as well as during pollution episodes. This approach effectively integrates observations with model simulations and emission inventories.
However, it is noted that this work appears very similar to a study published in 2022 by the same group (Popovicheva et al., 2022), conducted at the same location with similar observations and using the same model. The conclusions are largely the same, apart from the inclusion of two additional years of observations.
**Response**: The purpose of this paper was to present an extensive dataset that spans for 3.5 years instead of 1 year in Popovicheva et al. (2022). The large amount of research results collected during all these years helps us to verify previous observations and present new findings for winter/summertime maxima and the impact of the increasing wildfire events in Siberia. In addition, we have a set of new data on light absorption, such as the spectral dependence of light absorption, multi-annual intercomparison analyses between polar stations according with respect to babs(570) characteristics, analyses of 170 samples for OC/EC in order to virify correlations between EC and eBC (this has never done before), revise the eBC$_{AET}$ values and get site-specific mass absorption characteristics that quantify the light absorption for a given location (that is the most important and representative characteristic).

I would recommend publishing the manuscript in Atmospheric Chemistry and Physics (ACP) after the authors address the following aspects:

- Reduce or condense repetitive content that overlaps with the previous paper published in 2022.

**Response**: We have tried to reduce repetitive content in the new version. However, it should be noted that since we have extended the dataset, some repetition cannot be avoided.

- Emphasize aspects that are being discussed for the first time in this study, such as "BrC and its relationship with the corresponding AAE" and "eBC (derived from SAC) and its comparison with eBCAET".

**Response**: We appreciate Reviewer 2 for pointing this out and we have tried to focus on new findings as much as the results allow. Several changes have been performed throughout the manuscript in this aspect.

- It is noted that the eBC values are approximately half of the eBC$_{AET}$ values (Table 1). Could the authors have a further discussion regarding the difference?

**Response**: We have now added a discussion in section 3.2 at Lines 468-498 (see Manuscript with Track Changes).

- There appears to be a significant discrepancy between the results in Table S3 and Figure 10 b). For example, in July 2020, Figure 10 b) shows that biomass burning (BB) accounts for about 80%, whereas Table S3 indicates only a 7% contribution of BrC during the same period. Conversely, in February 2020, Figure 10 b) shows less than 10% BB, while Table S3 also reports a 7% BrC contribution. Could the authors clarify why these differences occur?

**Response**: Table S3 presents the contribution of babs from BrC to total babs(370) calculated from the aethalometer data. It relates to BrC contribution at 370 nm.
On the contrary, Figure 10b shows the source contributions to BC estimated by the FLEXPART model. Hence, the results in Table S3 and Figure 10 are not directly related.
Since we use BB emissions (wildfires) from CAMS GFAS (see Methods section), we can relate the modelled BB contribution to observations. Light absorption at 370 nm is considered as a "proxy" for BrC and can be related to BB (fires), when the BB sources are nearby. At IBS, as well as in other Arctic stations, our research indicates aged BBBC and, thus, comparison is not straightforward.

Specific comments:
L27-L28: Is the "92%" an average value of cold seasons over the entire study period? It appears that the 92% mentioned in abstract is not consistent with the 83% stated in the conclusion.
**Response**: We have tried to clarify (see Lines 28-30 in Manuscript with Track Changes). 83% is the contribution from anthropogenic sources over the entire period (including both cold and warm periods), 92% corresponds to the anthropogenic contribution during the cold season only, over for the 3.5 years of study).

L87-L88: Are the Biomass Burning (BB) results in Figure 10 a) and b) from ground level or from the altitudes (600-800 hPa)? Do the BB results in Figure 10 support the previously work by Qi and Wang, 2019?
**Response**: The Legend of Figure 10 states that "(a) Timeseries of monthly mean contribution from different emission source types to surface BC concentrations…". Hence, the answer is that the contribution corresponds to the surface (always below the boundary layer height).
Exactly! Our main findings support those reported by Qi & Wang (2019). We also find that the main source at IBS is fossil fuel combustion (e.g., gas flaring emissions), while wildfires play secondary role. However, here, we do not examine the impact of deposition on snow/ice, like in Qi & Wang (2019) (since we do not have such measurements). The impact of long-range transport from lower latitude Asian sources to the Arctic has been quantified long before by Evangeliou et al (2016) (https://doi.org/10.5194/acp-16-7587-2016).

L150: Should this be Fig. S1, instead of Fig.1d ?
**Response**: Correct! We have change this in Line 192 (see Manuscript with Track Changes).

L159: What are the two inlets with a much higher flow rate (~ 38LPM) than 5LPM?

**Response**: Information added in Lines 214-216 (see Manuscript with Track Changes).

L170: Please specify eBCAET as "equivalent black carbon concentration by aethalometer", when mentioning it for the first time.
**Response**: Corrected in Line 224 (see Manuscript with Track Changes).

L182: Please spell out for "NIR-VIS"
**Response**: Corrected in Line 236 (see Manuscript with Track Changes).

L240: What is the highest level of the 137 vertical levels?
**Response**: Approximately 80 km. We have added it in Line 317 (see Manuscript with Track Changes).

How do you couple with the BB burning injection height in the emission inventory?
**Response**: The BB emissions from CAMS-GFAS come with estimates of the maximum injection altitude of each gridded emission. We run our model backwards releasing particles at the surface of the receptor (this is the IBS site) and calculate footprint emission sensitivities at 0-100, 100-3000 and 3000-8000. We find the indices of the fires occurring at these heights, we calculate modelled concentrations at these heights separately and we sum them afterwards to retrieve the total modelled concentration at the receptor (IBS site).

L276-L277: Please be consistent with the wavelength mentioned (880nm or 800 nm?). Only 880nm is shown on Fig. 2.
**Response**: We appreciate Reviewer for pointing out this typo. The correct wavelength is 880 nm and has been now corrected everywhere in the manuscript. Please see section 3.1 and elsewhere in Manuscript with Track Changes.

L277: The expression is confusing, i.e., "the mean ± sigma (median) values". It is suggested to use "the mean ± sigma and the median, respectively".
**Response**: Corrected as reviewer suggested in Line 356 (see Manuscript with Track Changes).

L285: Please ensure consistency between the text and Table 2.
**Response**: There is no Table 2 in the manuscript, so we assume that Reviewer means consistency in terms of babs(880) (and not 800 as it was earlier). This has been corrected throughout the manuscript.

L295-L297: The sentence is not well expressed. Please consider re-phrasing it.
**Response**: We have removed this sentence as unnecessary, since it does not add anything to the manuscript.

L308 & L1083: Please ensure consistency in the wavelength between the text on page 11 and Figure 3(a). The caption for Figure 3(a) should indicate 590 nm. Additionally, please add (a) and (b) to Figure 3.
**Response**: We have corrected the inconsistency in legend of Figure 3. It now writes 590 nm (see Manuscript with Track Changes).

L378: Is it the cold season, or the warm season over the entire period?
**Response**: Line 378 writes "… observed for the cold and warm periods with means 44±47 and 19±57 ng m-3, respectively." We have now added ", for the entire study period." Please see Manuscript with Track Changes.

L385-L388: The numbers in the text differ slightly from those in Table S2. Please ensure consistency between the text and the table.
**Response**: We have corrected the number in consistency with Table S2 in Lines 540-544 (see Manuscript with Track Changes).

L427: No box-whisker plots are shown in Figure S3, which presents the emissions of BC from CAMS-GFAS. This should be Figure S4.
**Response**: Corrected in Line 595 (see Manuscript with Track Changes).

L431: It is suggested to add "in percentage" (shown in Table S3) after the "contribution".
**Response**: Corrected in Line 599 (see Manuscript with Track Changes).

L439: Can the authors explain why $eBC_{AET}$ is higher than eBC by a factor of two?
**Response**: We have now added this explanation in section 3.2 at Lines 468-498 (see Manuscript with Track Changes). This has been answered in a previous comment.

L498: Should it be Figure 9?
**Response**: Corrected in Line 683 (see Manuscript with Track Changes).

L520-L521: It is interesting to see a lack of consistency between Figure 10b and Table S3. It appears that a high BB contribution does not result in a relatively high percentage of BrC.
**Response**: Typically, a high BB contribution results in a high percentage of BrC in atmospheric aerosols. However, the exact relationship depends on several factors. What we know is that BrC is a major component of BB emissions and that BB produces both BC and BrC.
Although high BB contribution generally means more BrC, the actual percentage depends on (a) combustion conditions (flaming combustion – high temperature – results in more BC and less BrC, smoldering – low temperature – results in more BrC and less BC, peat fires – wet biomass – result in higher BrC), (b) fuel type (wood, agricultural waste, and peat produce high BrC emissions, coal burning – anthropogenic – results in more BC and less BrC), and (c) atmospheric aging (BrC undergoes photobleaching – degrades under sunlight – reducing its absorption over time and aged BrC may convert into non-absorbing organic carbon).

L546: It should be ~ 50%, instead of 60%.
**Response**: The correct number is 57% (see Line 741, Manuscript with Track Changes).

L547: It is suggested to use 72,400 km2.
**Response**: Corrected in Line 741, Manuscript with Track Changes.

L560: what is the unit of "150,000"?
**Response**: 150,000 "fires" occurred. Corrected now in Line 755 (see Manuscript with Track Changes).

L574: In general, the 'Conclusion' section is too long and should be more concise.
**Response**: The conclusions section has been shortened and refined as suggested by both reviewers (see Manuscript with Track Changes).

L596: "BrC light absorption coefficient in the UV spectrum showed similar trends as BC, although it exceeded BC by 2.4 times during both cold and warm periods." What is the basis of this statement?
**Response**: In Conclusions section we aimed to concentrate the basic findings of this work, considering that the paper is relatively long. In this sense, we simply report the findings. We have now changed the title of the section to Summary and conclusions.

L607: Please confirm the numbers. They should be consistent with these in Table S2, where the numbers are 106 ± 67 ng m-3.
**Response**: Corrected in Line 806 (see Manuscript with Track Changes). In total, we have tried once again to make text consistent with data show in Tables.

L669: Please use "MAC", instead of "MAC" to ensure consistency when writing the initials. Otherwise, it would be confused with the Mass Absorption Coefficient (MAC).
**Response**: Corrected in Line 939 (see Manuscript with Track Changes).

---

## Author Response (AR2)

**Editor decision: Publish subject to minor revisions (review by editor)**

by Andreas Petzold

**Public justification (visible to the public if the article is accepted and published)**:
Dear Nikolaos,

I am happy to accept the revised manuscript for publication in ACP after few minor revisions have been implemented.

1. The abstract exceeds the word count of 250 by about 40 words. Please shorten accordingly.

**Response**: The abstract was shortened now to 246 words (see Manuscript with Track Changes).

2. In line 217 you write "The aerosol pavilion takes place approximately half a km ...". I guess you mean "the aerosol pavilion is situated ... " If so, please rephrase.

**Response**: Corrected! Please check Line 182 (Manuscript with Track Changes).

3. in line 831 you write "Higher magnitude (around 4-5 times) in comparison with multi-year observations at high latitude polar stations in European Arctic ..." I guess you mean higher values of the light absorption coefficient. Please rephrase.

**Response**: Corrected! Please check Line 621 (Manuscript with Track Changes).

4. Please check all axis titles and units for readability in the printed version. For some of the figures the numbers at the axes seem very small for final size.

**Response**: We have tried to increase the fontsize in all axes to achieve better readability of the manuscript (please see manuscript with Track Changes). If this is not enough, I would guess we will receive another message, while in typesetting.

I am looking forward to the final version of your manuscript and will accept immediately after receipt and checking.

Best wishes
Andreas